# Generation then Reconstruction: Accelerating Masked Autoregressive Models via Two-Stage Sampling

**Feihong Yan** [1,2]    **Yao Zhu** [3]    **Peiru Wang** [2]    **Pang Kaiyu** [1]    **Qingyan Wei** [2]    **Huiqi Li** [✉1,4]
**Linfeng Zhang** [✉2]

[1] Beijing Institute of Technology    [2] Shanghai Jiao Tong University    [3] Zhejiang University
[4] Beijing Key Laboratory of Intelligent Diagnosis Technology and Equipment for Optic Nerve-Related Eye Diseases, Beijing, China    ✉ Corresponding authors
huiqili@bit.edu.cn    zhanglinfeng@sjtu.edu.cn

## Abstract

Masked Autoregressive (MAR) models promise better efficiency in visual generation than continuous autoregressive (AR) models for the ability of parallel generation, yet their acceleration potential remains constrained by the modeling complexity of spatially correlated visual tokens in a single step. To address this limitation, we introduce **Generation then Reconstruction** (GtR), a training-free hierarchical sampling strategy that decomposes generation into two stages: structure generation establishing global semantic scaffolding, followed by detail reconstruction efficiently completing remaining tokens. Assuming that it is more difficult to create an image from scratch than to complement images based on a basic image framework, GtR is designed to achieve acceleration by computing the reconstruction stage quickly while maintaining the generation quality by computing the generation stage slowly. Moreover, observing that tokens on the details of an image often carry more semantic information than tokens in the salient regions, we further propose **Frequency-Weighted Token Selection** (FTS) to offer more computation budget to tokens on image details, which are localized based on the energy of high frequency information. Extensive experiments on ImageNet class-conditional and text-to-image generation demonstrate $3.72\times$ speedup on MAR-H while maintaining comparable quality (*e.g.,* FID: 1.59, IS: 304.4 vs. original 1.59, 299.1), substantially outperforming existing acceleration methods across various model scales and generation tasks. Our codes will be released in https://github.com/feihongyan1/GtR.

## 1 Introduction

Motivated by the successes of autoregressive (AR) models in natural language processing, the realm of computer vision has increasingly explored the autoregressive paradigm for visual content generation (Van Den Oord et al., 2016; Chen et al., 2020; Yu et al., 2022; Tian et al., 2024; Zhao et al., 2025a). Early endeavors adopt pixel-by-pixel generation strategies(Van den Oord et al., 2016), treat images as flattened sequences, and apply causal modeling directly. However, the autoregressive formulation suffers from severe computational inefficiency, due to the natural inability to the parallel generation. To solve this problem, an alternative direction emerges through next-set prediction, exemplified by Masked Autoregressive (MAR) models (Chang et al., 2022; Li et al., 2023; 2024). MARs adopt an encoder-decoder architecture with bidirectional attention, where the encoder produces conditioning vectors $z$ for each token, subsequently guiding a diffusion process to generate the final tokens. This framework enables simultaneous prediction of multiple tokens in a single forward pass while maintaining competitive generation quality.

Although parallelism has been provided by MAR, directly generating too many tokens in a single step usually brings a significant degradation in the quality of generation in practice. Concretely, this problem arises from the inherent complexity of modeling high-dimensional joint distributions. Visual

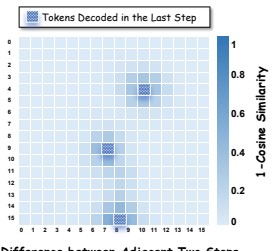

Figure 1: **The difference of token features in the adjacent steps.** Once a token is decoded, its adjacent tokens tend to exhibit significant changes.

Figure 2: **Correspondence between high-frequency regions in pixel space and feature space.** Each triple shows three images: original image, pixel-space high-frequency heatmap via high-pass filtering, and frequency heatmap of MAR's conditioning vectors. The spatial alignment demonstrates that high-frequency tokens in feature space indicate regions with fine-grained textures and high-frequency details.

tokens exhibit strong spatial correlations that fundamentally violate conditional independence assumptions, necessitating explicit modeling of interdependencies through joint probability distributions rather than simplified factorizations. When simultaneously predicting multiple tokens, MARs must estimate the joint probability distribution over all target tokens whose modeling difficulty increases with the number of predicted tokens, limiting the speed of parallel generation. To solve this problem, we begin by exploring the intrinsic property of MAR.

**The spatially adjacent tokens tend to influence each other.** It is well-acknowledged that tokens in the same image region usually share similar semantic information. As a result, the spatially adjacent tokens tend to influence each other. Figure 1 shows the difference between tokens in the adjacent steps, which demonstrates that when one token has been decoded, its adjacent tokens tend to be significantly influenced, indicating that *adjacent tokens should be decoded separately*.

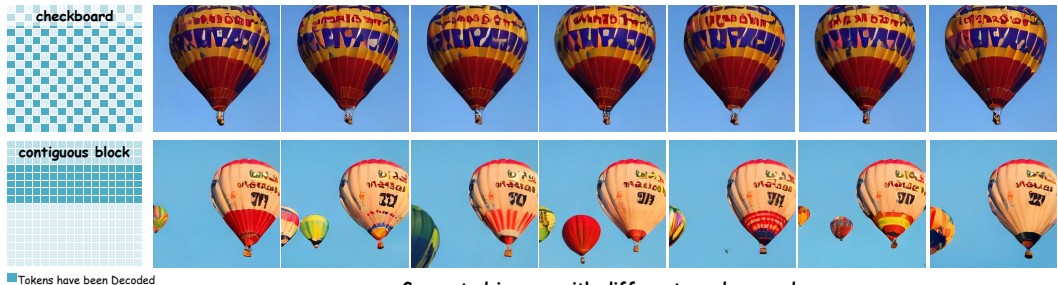

Figure 3: **Comparison of generation consistency in two sampling methods when 50% tokens have been generated and the remaining tokens are generated using different random seeds.** Dark blue and light blue indicate the tokens that have been decoded and not decoded, respectively. The top row shows checkerboard pattern that distributes generated tokens (dark blue) uniformly throughout the image, yielding consistent generation results. The bottom row shows contiguous block pattern that concentrates generated tokens in the upper region, resulting in a diverse generation.

**Covering more spatial locations indicates creating more information.** Figure 3 compares the generation results of two different sampling orders, including *"checkerboard"*, where the decoded tokens are isolated and spatially uniformly distributed, and *"contiguous block"*, where the tokens in the upper regions are generated. For each sampling order, we first decode the same 50% tokens, and then decode the left 50% tokens with different random seeds. Interestingly, we find that the seven images generated by "checkerboard" are almost identical, while the seven images generated by "contiguous block" exhibit a significant difference, indicating that the content of images has been almost fully decided when the generated tokens are spatially uniformly decided. *Concretely, decoding the tokens that cover most spatial locations has already "created" the main body of the image, while generating the left tokens is more likely to be an image "reconstruction" which does not bring new information and is much easier than "creation".*

Based on the two observations, we propose GtR (Generation-then-Reconstruction), which introduces a two-stage checkerboard-style generation process. In the first stage, we randomly generate tokens in the light blue positions in the checkerboard, which guarantees that the decoded tokens are not closely spatially adjacent and can cover most spatial positions in the images. As a result, this stage "creates" the main semantic content of the image, and thus it is performed at a slower speed (*e.g., generating fewer tokens at each step*). Then, in the second stage, since "reconstruction" is much easier than "creation", GtR introduces a highly parallel generation, which decodes all the left tokens in very few steps with a high parallel ratio, which can even be performed by a single step, thus bringing extreme acceleration without loss of generation quality.

Besides, the computation process of MAR includes not only the encoder and decoder, but also a diffusion model which maps the latent of each decoded token into a continuous vector, which also accounts for noticeable computation costs. The original MAR pays the same computation costs for each token, while ignoring the fact that the token with great details and complex patterns is much more difficult to generate. In this paper, we further propose Frequency-Weighted Token Selection (FTS), a training-free strategy that allocates more diffusion steps to the tokens with more details. As demonstrated in Figure 2, FTS applies a Fourier transformation to the latents of tokens, and then identifies the tokens with larger high-frequency energy as the tokens with more details.

## 2 RELATED WORK

**Next-Token Autoregressive Visual Generation.** Autoregressive models have been increasingly adopted across a wide range of domains, including natural language processing, visual understanding and generation (Liu et al., 2025b; 2026; 2025a; Zhao et al., 2022). Autoregressive models for visual generation must map two-dimensional image structures into sequential one-dimensional token representations. Early studies explored RGB pixel synthesis via row-by-row raster-scan (Chen et al., 2020; Gregor et al., 2014; Van Den Oord et al., 2016; Van den Oord et al., 2016) methodology. VQGAN (Esser et al., 2021) establishes the foundation by converting two-dimensional visual content into one-dimensional discrete token sequences, while VQVAE-2 (Razavi et al., 2019) and RQ-Transformer (Lee et al., 2022) extend it with hierarchical or stacked representations. Building on this foundation, LlamaGen (Sun et al., 2024) scales the architecture to 3B parameters on the LLaMA (Touvron et al., 2023) framework, achieving quality comparable to competitive diffusion models. Recent advances extend autoregressive generation beyond fixed raster-scan orders. RAR (Yu et al., 2024), DAR (Xu et al., 2025), and D-AR (Gao & Shou, 2025) introduce randomized, diagonal, or diffusion-inspired factorization schemes, while FractalGen (Li et al., 2025) expand token representations through flexible entities and fractal composition, thereby enhancing modeling flexibility and performance. Nevertheless, next-token-prediction paradigms face a fundamental computational bottleneck: their strictly sequential nature allows only one token per inference step, and the long token sequences of images make real-time generation prohibitive.

**Next-Scale Visual Generation.** Multi-scale generation offers an alternative to mitigate the computational cost of token-by-token prediction. MUSE (Chang et al., 2023) hierarchically generates low-resolution tokens with a base transformer and then SuperRes is used to generate high-resolution ones. Hi-MAR (Zheng et al., 2025b) further uses a unified masked autoregressive model that first predicts low-resolution token pivots to capture global structure, then conditions on them to generate the full-resolution image. However, it introduces additional KV cache overhead and necessitates expensive, unstable training with a specialized multi-scale loss. VAR (Tian et al., 2024) adopts a decoder-only transformer configuration for next-scale prediction, which reduces computational overhead and improves scalability. In addition, E-CAR (Yuan et al., 2024) and NFIG (Huang et al., 2025) adopt a coarse-to-fine strategy by multistage generation in continuous token space and using frequency-aware decomposition respectively. CTF (Guo et al., 2025) mitigates quantization redundancy by autoregressively predicting coarse labels and refining them in parallel. However, this multi-scale tokenization methodology exhibits fundamental incompatibility with the 1D flat token representation paradigm that has been extensively integrated into contemporary multimodal systems, potentially limiting its broader applicability.

**Next Set-of-Tokens Visual Generation.** Non-autoregressive generation methods have emerged as a promising alternative to sequential token prediction. MaskGIT (Chang et al., 2022) pioneered the

next set-of-tokens prediction paradigm, leveraging BERT-style (Devlin et al., 2019; Wei et al., 2025) bidirectional attention mechanisms and enabling parallel replacement of multiple masked tokens through stochastic sampling or confidence-based selection strategies. MAR (Li et al., 2024) extends MaskGIT's framework by introducing diffusion-based loss functions to transform from discrete token representations to continuous token spaces, thereby mitigating information loss. Building on this paradigm, ZipAR (He et al., 2025a), NAR (He et al., 2025b), and Harmon (Wu et al., 2025a) exemplify efforts to relax strict sequential prediction, either through spatially localized or neighboring-token parallelism, or by unifying visual understanding and generation within a shared masked autoregressive framework. xAR (Ren et al., 2025a) predicts coarse-grained units (e.g., cell of tokens) per step, enabling parallel intra-unit generation while maintaining inter-unit autoregressive dependencies. Although next set-of-tokens prediction reduces sampling iterations, MAR remains limited in parallelization and requires computation across all token positions, leading to suboptimal efficiency. Recent methods have attempted to accelerate next set-of-tokens prediction. LazyMAR (Yan et al., 2025) accelerates MARs through token and condition caching mechanisms, but it does not improve the sampling strategy of MARs. DiSA (Zhao et al., 2025b) proposes a diffusion step annealing strategy for the diffusion head in MARs, but it overlooks the differences in modeling difficulty among different image regions. Halton-MaskGIT (Besnier et al., 2025) introduces the Halton scheduler into MaskGIT, but its fixed ordering reduces diversity in image generation.

## 3 METHOD

### 3.1 PRELIMINARY

**Image Generation as Next-Token Prediction.** With an image tokenizer, the input image $\mathbf{I} \in \mathbb{R}^{H \times W \times 3}$ is then encoded into $h \times w$ tokens, where $h = H/p$, $w = W/p$, and $p$ denotes the downsampling ratio. These tokens are reshaped into a sequence $\mathbf{x} = (x^1, x^2, x^3, \ldots, x^n)$ with $n = h \cdot w$, arranged in raster scan order. The joint distribution is factorized as $p(\mathbf{x}) = \prod_{i=1}^{n} p\left(x^i \mid x^1, x^2, \ldots, x^{i-1}\right)$, where $p\left(x^i \mid x^1, x^2, \ldots, x^{i-1}\right)$ represents the conditional distribution of token $x^i$ given previous tokens $x^1$ to $x^{i-1}$. However, raster-order prediction cannot capture overall image structure early in generation, and sequential processing scales linearly with resolution.

**Image Generation as Next-Set Prediction.** To overcome the sequential bottleneck, next-set prediction enables simultaneous generation of multiple tokens within each inference step. Let $\tau$ denote a random permutation of $[1, 2, \ldots, n]$. The joint distribution is decomposed into $N$ prediction steps:

$$p(x^1, ..., x^n) = \prod_{k=1}^{N} p(X^k \mid X^1, ..., X^{k-1}) \tag{1}$$

where $X^k = \{x^{\tau_i}, x^{\tau_{i+1}}, \ldots, x^{\tau_j}\}$ represents the $k$-th token subset under permutation $\tau$, with constraints $\bigcup_{k=1}^{N} X^k = \{x^1, \ldots, x^n\}$ and $X^i \cap X^j = \emptyset$ for $i \neq j$. MARs rewrite this formulation in two parts: generating conditioning vectors $z^k = f(X^1, \ldots, X^{k-1})$ via bidirectional attention, then modeling $p(X^k \mid z^k)$ through diffusion, enabling higher-quality continuous-valued token generation.

**Limitations of MARs.** MARs exhibit two fundamental limitations: (1) *Spatial correlation modeling*: Random permutation may simultaneously predict spatially adjacent tokens, which is more challenging than predicting spatially separated tokens. (2) *Violation of composition-to-detail paradigm*: Humans typically perceive and create visual content hierarchically, first establishing global structure then refining local details. However, random token sampling violates this paradigm and may create blank areas in later generation stages, degrading quality due to insufficient context.

### 3.2 GENERATION THEN RECONSTRUCTION

To address the limitations of MARs, we propose **GtR (Generation-then-Reconstruction)**, which introduces a two-stage checkerboard-style generation process that decomposes visual generation into semantic creation followed by detail reconstruction. As illustrated in Figure 4, given an image tokenized into $h \times w$ tokens, let $i, j$ represent the row and column indices of each token position, the generation stage randomly generates tokens where $(i + j) \bmod 2 = 0$ at a slower speed (e.g., generating fewer tokens at each masked autoregressive step) to establish the main semantic structure,

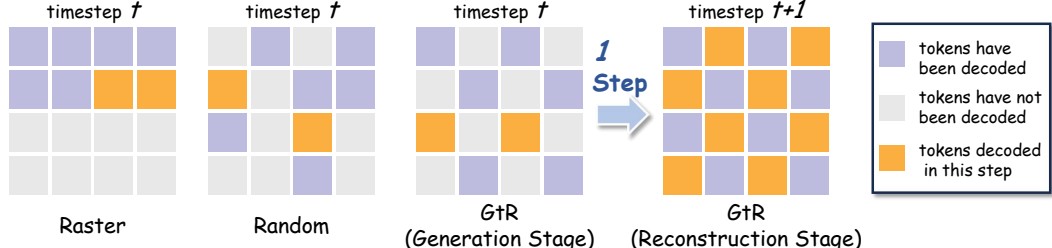

Figure 4: **Comparison of different token sampling strategies.** The proposed GtR formulates the generation process as a two-stage checkerboard procedure: generation stage establishes global semantic structure through spatially non-adjacent tokens at conservative speed, followed by reconstruction stage completing remaining tokens within 1-2 steps through highly parallel generation.

while the reconstruction stage subsequently generates the remaining tokens where $(i + j) \bmod 2 = 1$ in very few steps with a high parallel ratio, which can even be performed by a single step to bring extreme acceleration without loss of generation quality.

---

**Algorithm 1** Stage Partitioning

---

**Require:** resolution $h \times w$, number of stages $K$, initial stack $\mathcal{T}$ and token set $\mathcal{R} = \{(i, j) : 0 \leq i < h, 0 \leq j < w\}$
  1: **for** $k = 1$ to $K - 1$ **do**
  2:    $\mathcal{U}_k \leftarrow \{(i, j) \in \mathcal{R} : (i + j) \bmod 2^k = 2^{k-1}\}$       ▷ Extract tokens with remainder $2^{k-1}$
  3:    Push $\mathcal{U}_k$ onto stack $\mathcal{T}$
  4:    $\mathcal{R} \leftarrow \{(i, j) \in \mathcal{R} : (i + j) \bmod 2^k = 0\}$       ▷ Update remaining tokens with remainder 0
  5: **end for**
  6: Push $\mathcal{R}$ onto stack $\mathcal{T}$
  7: $\{\mathcal{S}_k\}_{k=1}^{K} \leftarrow$ Pop all elements from stack $\mathcal{T}$ in LIFO order
  8: **return** stage partitions $\{\mathcal{S}_k\}_{k=1}^{K}$

---

However, the simple two-stage framework may suffer from delayed semantic structure establishment, as randomly sampling tokens within the generation stage could concentrate generated tokens in localized regions, thereby postponing the formation of global semantic guidance until later steps. To address this limitation, we further subdivide the generation stage into $K - 1$ sub-stages, which enables the first sub-stage to generate spatially uniform tokens distributed across the entire image in fewer masked autoregressive steps, thereby establishing the fundamental semantic structure as soon as possible and providing robust conditioning for subsequent generation.

Algorithm 1 partitions the complete token set into $K$ disjoint subsets $\mathcal{S} = \{\mathcal{S}_1, \mathcal{S}_2, \ldots, \mathcal{S}_K\}$, which are allocated to the $K - 1$ sub-stages of the generation stage and the reconstruction stage, where $\bigcup_{k=1}^{K} \mathcal{S}_k = \{x^1, \ldots, x^n\}$ and $\mathcal{S}_i \cap \mathcal{S}_j = \emptyset$ for $i \neq j$. Algorithm 1 iteratively bisects the unassigned token set $\mathcal{R}$ into two subsets at each iteration: one subset is allocated to a new sub-stage, while the other becomes the updated $\mathcal{R}$ for the subsequent iteration. This hierarchical decomposition ensures that tokens within each subset are uniformly distributed throughout the image, effectively increasing the spatial distance between simultaneously predicted tokens and reducing their interdependence. As more tokens are generated, the fundamental semantic structure of the image becomes increasingly established, providing stronger conditioning for subsequent token prediction and enabling later stages $k$ to achieve higher generation rates $r_k$.

Tokens generated in later stages are conditioned on the tokens from all previous stages. A causal dependency is formed between these stages. Consequently, the joint distribution of all tokens can be reformulated as follows:

$$p(x^1, \ldots, x^n) = \prod_{k=1}^{K} p(\mathcal{S}_k \mid \mathcal{S}_1, \ldots, \mathcal{S}_{k-1}) \tag{2}$$

where $p(\mathcal{S}_k \mid \mathcal{S}_1, \ldots, \mathcal{S}_{k-1})$ represents the conditional distribution of tokens in stage $k$ given all tokens generated in the previous stages $\mathcal{S}_1$ through $\mathcal{S}_{k-1}$. After initial structure generation, the checkerboard pattern ensures that each ungenerated token is surrounded by generated tokens. This forms strong causal dependencies where ungenerated tokens are directly conditioned on their neighboring generated tokens, which constrains the token distributions and enables the remaining half of image tokens to be generated within as few as 1 to 2 masked autoregressive steps.

**Intra-Stage Masked Generation.** Within each stage $k$, we generate all tokens $\mathcal{S}_k$ through $M_k$ masked autoregressive steps, where $M_k \leq |\mathcal{S}_k|$. At each masked autoregressive step $m$ within stage $k$, we use next-set prediction to sample a subset of tokens $X^{k,m} \subseteq \mathcal{S}_k$, conditioning on all tokens from previous stages and the tokens already generated within the current stage. Formally, the probability of generating tokens in stage $k$ is decomposed as follows:

$$p(\mathcal{S}_k \mid \mathcal{S}_{<k}) = \prod_{m=1}^{M_k} p(X^{k,m} \mid X^{k,1}, \ldots, X^{k,m-1}, \mathcal{S}_{<k}) \tag{3}$$

where $\mathcal{S}_{<k} = \bigcup_{i=1}^{k-1} \mathcal{S}_i$ represents all tokens generated in previous stages, and $\bigcup_{m=1}^{M_k} X^{k,m} = \mathcal{S}_k$ with $X^{k,i} \cap X^{k,j} = \emptyset$ for $i \neq j$. The likelihood of the complete token sequence $\mathbf{x} = (x^1, x^2, x^3, \ldots, x^n)$ can be reformulated as follows:

$$p(x^1, \ldots, x^n) = \prod_{k=1}^{K} \prod_{m=1}^{M_k} p(X^{k,m} \mid X^{k,1}, \ldots, X^{k,m-1}, \mathcal{S}_{<k}) \tag{4}$$

Equation 4 can be viewed as a reformulation of Equation 1. Our method still follows the next-set prediction paradigm, differing only in sampling order. Because MARs are trained on random permutations of all possible token orders, including the sampling order of GtR, our method can be applied to any MAR models in a training-free manner.

**Stage-Aware Diffusion Scheduling.** The computational process of MARs includes not only the encoder and decoder, but also a diffusion model for modeling the per-token probability distribution. However, traditional MARs apply the same diffusion steps to each masked autoregressive step, ignoring the changes in modeling complexity across different masked autoregressive steps. As our first sub-stage of the generation stage establishes fundamental semantic structure through spatially distributed tokens, subsequent generation is guided by more accumulated conditional information and becomes easier. Therefore, we implement linearly decreasing diffusion steps from $T_{\max}$ to $T_{\min}$ during the generation stage and set the diffusion steps to $T_{\mathrm{rec}}$ throughout the reconstruction stage.

### 3.3 FREQUENCY-WEIGHTED TOKEN SELECTION

During the reconstruction stage, tokens exhibit heterogeneous prediction complexity, and tokens corresponding to regions with complex and fine textures are difficult to accurately model with $T_{\mathrm{rec}}$ diffusion steps. To address this limitation, we propose Frequency-Weighted Token Selection (FTS) that identifies structurally critical tokens and allocates additional diffusion steps accordingly.

Let $\mathbf{z}^i \in \mathbb{R}^D$ denote the conditioning feature produced by the autoregressive model for token $x^i$, where $D$ represents the feature dimensionality. To analyze the frequency characteristics of these conditioning features, we apply the Discrete Fourier Transform to each token's conditioning vector: $\mathcal{F}\left(\mathbf{z}^i\right)(n) = \sum_{d=0}^{D-1} \mathbf{z}^i(d) \cdot e^{-j\frac{2\pi n d}{D}}, \quad n = 0, 1, \ldots, \lfloor D/2 \rfloor$ where $\mathbf{z}^i(d)$ denotes the $d$-th element of the conditioning feature. The amplitude spectrum is computed from the real and imaginary components of the Fourier transform: $\mathcal{A}\left(\mathbf{z}^i\right)(n) = \left[R^2\left(\mathcal{F}\left(\mathbf{z}^i\right)(n)\right) + I^2\left(\mathcal{F}\left(\mathbf{z}^i\right)(n)\right)\right]^{1/2}$, where $R(\cdot)$ and $I(\cdot)$ represent the real and imaginary parts of the complex Fourier coefficients, respectively.

The importance score for each token is computed through weighted integration of its frequency spectrum, where higher frequency components receive linearly increasing weights:

$$s^i = \sum_{n=1}^{\lfloor D/2 \rfloor} \mathcal{A}\left(\mathbf{z}^i\right)(n) \cdot \left(1 + \frac{n}{\lfloor D/2 \rfloor}\right) \tag{5}$$

We rank all tokens by their importance scores $s^i$ and assign $T_{\mathrm{detail}}$ diffusion steps to the top $\beta$ high-frequency tokens during the reconstruction stage to model complex texture regions more accurately.

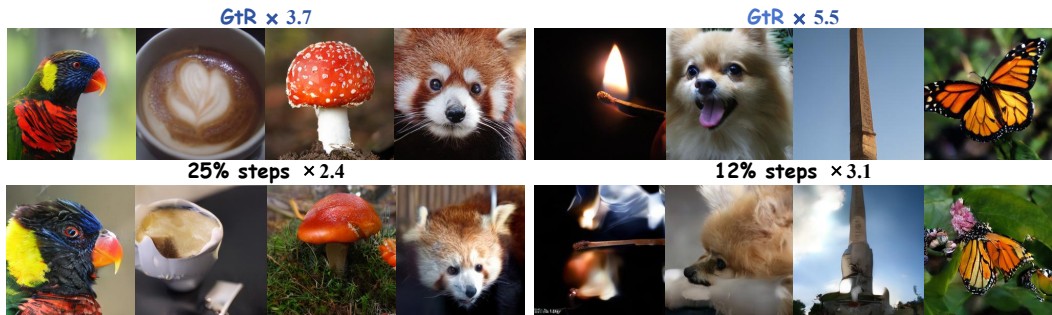

Figure 5: **Qualitative comparison** between Generation then Reconstruction (GtR) and the acceleration achieved through step reduction on MAR. GtR enables extreme acceleration while maintaining generation quality, whereas step reduction results in significant visual degradation.

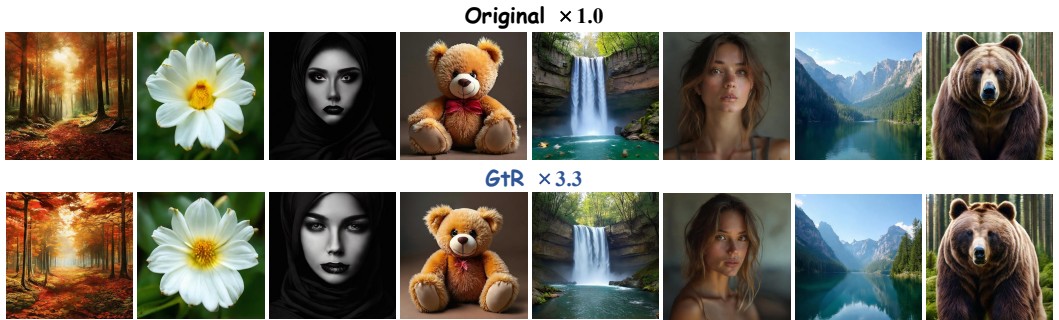

Figure 6: **Qualitative comparison** of generation results between GtR and the original LightGen. GtR achieved a $3.3\times$ speedup while maintaining generation quality comparable to the original LightGen.

## 4 EXPERIMENTS

### 4.1 EXPERIMENTAL SETTINGS

**Implementation Details.** We evaluate our method on two generation tasks: (1) class-conditional ImageNet generation using MAR (Li et al., 2024) variants (MAR-B/L/H with 208M/479M/943M parameters) at $256 \times 256$ resolution, where images are tokenized into 256 tokens via KL-16 tokenizer (Rombach et al., 2022) with 100 diffusion steps following LazyMAR (Yan et al., 2025); (2) text-to-image generation using 7B LightGen (Wu et al., 2025b) at $512 \times 512$ resolution, where images are decomposed into 1024 tokens via VAE encoder with 50 diffusion steps. Text prompts are encoded through T5-XXL (Raffel et al., 2020). For GtR implementation, we use $K = 3$ stages for MAR with generation rates $r_k = \{2.67, 10.67, 64\}$ and $K = 4$ stages for LightGen with $r_k = \{16, 42.6, 85.3, 256\}$. Stage-aware diffusion scheduling employs linearly decreasing steps from $T_{\max} = 50$ to $T_{\min} = 20$ during generation stages, with $T_{\text{rec}} = 20$ during reconstruction. FTS allocates $T_{\text{detail}} = 50$ diffusion steps to the top $\beta = 10\%$ high-frequency tokens.

**Evaluation Metrics.** For MAR, we generate 50,000 images across the 1,000 classes of ImageNet-1K and evaluate image quality using FID (Heusel et al., 2017) and IS (Salimans et al., 2016) as standard metrics. We measure computational efficiency through FLOPs, CPU latency, and GPU latency. For LightGen, we use GenEval (Ghosh et al., 2023) to evaluate image generation quality.

### 4.2 CLASS-CONDITIONAL IMAGE GENERATION

We evaluate our method by comparing with the original MAR model, other MAR acceleration methods (Yan et al., 2025; Zhao et al., 2025b; Besnier et al., 2025), and state-of-the-art image generation models (Zheng et al., 2025a; Besnier et al., 2025; Sun et al., 2024; Ren et al., 2025b; Li et al., 2023; Peebles & Xie, 2023). As shown in Table 1, it can be observed that: (1) Compared to other

Table 1: **Model comparison results** on ImageNet $256 \times 256$ class-conditional generation. "MAR-B, -L, -H" denote MAR's base, large, and huge models. "64, 16" represent the number of decoding steps.

| Method | Inference Efficiency | | | | | Generation Quality | |
|---|---|---|---|---|---|---|---|
| | Latency(GPU/s)↓ | Latency(CPU/s)↓ | FLOPs(T)↓ | Speed↑ | Param | FID↓ | IS↑ |
| MAGE | 1.60 | 12.60 | 4.19 | 1.00 | 307M | 6.93 | 195.8 |
| LDM-4 | 5.35 | 27.25 | 69.50 | 1.00 | 400M | 3.60 | 247.7 |
| DiT-XL/2 | 4.84 | 196.88 | 114.38 | 1.00 | 675M | 2.27 | 278.2 |
| LlamaGen-3B | 1.65 | 1524.63 | 7.01 | 1.00 | 3.1B | 3.05 | 222.3 |
| Halton-MaskGIT | 0.46 | 20.13 | 10.72 | 1.00 | 705M | 4.17 | 263.0 |
| Hita-2B | - | - | - | - | 2B | 2.59 | 281.9 |
| FlowAR-H | 0.51 | 60.27 | 38.43 | 1.00 | 1.9B | 2.65 | 296.5 |
| MAR-B (64) | 0.26 | 27.83 | 14.49 | 1.00 | 208M | 2.32 | 281.1 |
| +Step=16 | $0.11_{-0.15}$ | $12.16_{-15.67}$ | $6.22_{-8.27}$ | 2.33 | 208M | $4.10_{+1.78}$ | $247.5_{-33.6}$ |
| +Halton | $0.11_{-0.15}$ | $12.16_{-15.67}$ | $6.22_{-8.27}$ | 2.33 | 208M | $3.37_{+1.05}$ | $257.1_{-24.0}$ |
| +DiSA | $0.09_{-0.17}$ | $8.78_{-19.05}$ | $4.82_{-9.67}$ | 3.01 | 208M | $2.52_{+0.20}$ | $272.9_{-8.2}$ |
| +LazyMAR | $0.09_{-0.17}$ | $8.19_{-19.64}$ | $4.11_{-10.38}$ | 3.53 | 208M | $2.64_{+0.32}$ | $276.0_{-5.1}$ |
| +GtR (Ours) | $0.07_{-0.19}$ | $7.18_{-20.65}$ | $3.87_{-10.62}$ | 3.74 | 208M | $2.37_{+0.05}$ | $283.5_{+2.4}$ |
| +Step=8 | $0.08_{-0.18}$ | $8.65_{-19.18}$ | $4.84_{-9.65}$ | 2.99 | 208M | $13.12_{+10.80}$ | $180.9_{-100.2}$ |
| +Halton | $0.08_{-0.18}$ | $8.65_{-19.18}$ | $4.84_{-9.65}$ | 2.99 | 208M | $9.44_{+7.12}$ | $204.6_{-76.5}$ |
| +DiSA | $0.06_{-0.20}$ | $6.52_{-21.31}$ | $3.46_{-11.03}$ | 4.19 | 208M | $3.62_{+1.30}$ | $255.7_{-25.4}$ |
| +LazyMAR | $0.06_{-0.20}$ | $5.49_{-22.34}$ | $2.70_{-11.79}$ | 5.37 | 208M | $4.37_{+2.05}$ | $241.9_{-39.2}$ |
| +GtR (Ours) | $0.05_{-0.21}$ | $4.30_{-23.53}$ | $2.29_{-12.20}$ | 6.33 | 208M | $2.76_{+0.44}$ | $274.6_{-6.5}$ |
| MAR-L (64) | 0.48 | 55.62 | 32.75 | 1.00 | 479M | 1.82 | 296.1 |
| +Step=16 | $0.19_{-0.29}$ | $21.92_{-33.70}$ | $13.42_{-19.33}$ | 2.44 | 479M | $4.32_{+2.50}$ | $247.4_{-48.7}$ |
| +Halton | $0.19_{-0.29}$ | $21.92_{-33.70}$ | $13.42_{-19.33}$ | 2.44 | 479M | $3.24_{+1.42}$ | $261.1_{-35.0}$ |
| +DiSA | $0.16_{-0.32}$ | $18.39_{-37.23}$ | $11.03_{-21.72}$ | 2.97 | 479M | $2.23_{+0.41}$ | $281.1_{-15.0}$ |
| +LazyMAR | $0.16_{-0.32}$ | $17.01_{-38.61}$ | $9.35_{-23.40}$ | 3.50 | 479M | $2.11_{+0.29}$ | $284.4_{-11.7}$ |
| +GtR (Ours) | $0.13_{-0.35}$ | $14.98_{-40.64}$ | $8.85_{-23.90}$ | 3.71 | 479M | $1.81_{-0.01}$ | $297.4_{+1.3}$ |
| +Step=8 | $0.14_{-0.34}$ | $15.87_{-39.75}$ | $10.21_{-22.54}$ | 3.21 | 479M | $16.11_{+14.29}$ | $165.0_{-131.1}$ |
| +Halton | $0.14_{-0.34}$ | $15.87_{-39.75}$ | $10.21_{-22.54}$ | 3.21 | 479M | $14.16_{+12.34}$ | $155.4_{-140.7}$ |
| +DiSA | $0.12_{-0.36}$ | $13.19_{-42.43}$ | $7.85_{-24.90}$ | 4.17 | 479M | $3.86_{+2.04}$ | $254.5_{-41.6}$ |
| +LazyMAR | $0.11_{-0.37}$ | $10.81_{-44.81}$ | $5.77_{-26.98}$ | 5.37 | 479M | $4.07_{+2.25}$ | $247.9_{-48.2}$ |
| +GtR (Ours) | $0.08_{-0.40}$ | $8.75_{-46.87}$ | $5.18_{-27.57}$ | 6.32 | 479M | $2.33_{+0.51}$ | $281.5_{-14.6}$ |
| MAR-H (64) | 0.81 | 104.66 | 64.52 | 1.00 | 943M | 1.59 | 299.1 |
| +Step=16 | $0.33_{-0.48}$ | $43.39_{-61.27}$ | $27.11_{-37.41}$ | 2.38 | 943M | $4.49_{+2.90}$ | $242.9_{-56.2}$ |
| +Halton | $0.33_{-0.48}$ | $43.39_{-61.27}$ | $27.11_{-37.41}$ | 2.38 | 943M | $3.18_{+1.59}$ | $261.7_{-37.4}$ |
| +DiSA | $0.27_{-0.54}$ | $33.72_{-70.94}$ | $21.59_{-42.93}$ | 2.99 | 943M | $2.11_{+0.52}$ | $283.1_{-16.0}$ |
| +LazyMAR | $0.27_{-0.54}$ | $32.10_{-72.56}$ | $18.85_{-45.67}$ | 3.42 | 943M | $1.94_{+0.35}$ | $284.1_{-15.0}$ |
| +GtR (Ours) | $0.22_{-0.59}$ | $27.93_{-76.73}$ | $17.34_{-47.18}$ | 3.72 | 943M | $1.59_{+0.00}$ | $304.4_{+5.3}$ |
| +Step=8 | $0.26_{-0.55}$ | $31.52_{-73.14}$ | $20.88_{-43.64}$ | 3.09 | 943M | $17.66_{+16.07}$ | $158.0_{-141.1}$ |
| +Halton | $0.26_{-0.55}$ | $31.52_{-73.14}$ | $20.88_{-43.64}$ | 3.09 | 943M | $11.85_{+10.26}$ | $191.2_{-107.9}$ |
| +DiSA | $0.19_{-0.62}$ | $24.10_{-80.56}$ | $15.44_{-49.08}$ | 4.18 | 943M | $3.15_{+1.56}$ | $265.5_{-33.6}$ |
| +LazyMAR | $0.18_{-0.63}$ | $21.28_{-83.38}$ | $12.74_{-51.78}$ | 5.06 | 943M | $4.06_{+2.47}$ | $249.3_{-49.8}$ |
| +GtR (Ours) | $0.15_{-0.66}$ | $19.98_{-84.68}$ | $10.27_{-54.25}$ | 6.28 | 943M | $2.16_{+0.57}$ | $285.6_{-13.5}$ |

state-of-the-art image generation models, MAR-H + GtR maintains the lowest GPU latency while achieving the best generation results. (2) Compared with the original MAR, our method achieves a $3.72\times$ speedup while maintaining nearly identical generation quality. Additionally, both MAR-H + GtR and MAR-L + GtR simultaneously surpass the original MAR's smaller variants in both quality and efficiency. (3) Our method outperforms other MAR acceleration methods (HaltonMAR, DiSA, and LazyMAR) in both speedup and generation quality. Even at extreme acceleration ratios, while other MAR acceleration methods exhibit significant degradation in generation quality, our method maintains comparable visual fidelity as demonstrated in Figure 5.

## 4.3 TEXT-TO-IMAGE GENERATION

We evaluate the acceleration performance of GenEval at $512 \times 512$ resolution and compare it with the original LightGen (Wu et al., 2025b) and other text-to-image Generation models (Rombach et al., 2022; Podell et al., 2024). As shown in Table 2, GtR achieves higher acceleration ratios compared to the original model and LazyMAR, while simultaneously delivering superior generation quality. Figure 6 illustrates the results under a $3.3\times$ acceleration setting with GtR, where the generated images remain largely consistent with the original outputs.

Table 2: **Performance comparison in 512 × 512 on GenEval.** GtR achieves 3.82× speedup while maintaining superior generation quality compared to baseline LightGen.

| Methods | Inference Efficiency | | | Generation Quality | | | | | | |
|---|---|---|---|---|---|---|---|---|---|---|
| | Latency(GPU/s)↓ | Speed↑ | Param↓ | Single Obj.↑ | Two Obj.↑ | Colors↑ | Counting↑ | Position↑ | Color Attri.↑ | Overall↑ |
| SDv1.5 | 0.97 | 1.00 | 0.9B | 0.96 | 0.38 | 0.77 | 0.37 | 0.03 | 0.05 | 0.42 |
| SDv2.1 | 0.87 | 1.00 | 0.9B | 0.91 | 0.24 | 0.69 | 0.14 | 0.03 | 0.06 | 0.34 |
| SDXL | 1.46 | 1.00 | 2.6B | 0.63 | 0.23 | 0.51 | 0.12 | 0.04 | 0.05 | 0.26 |
| Llamagen | 3.13 | 1.00 | 0.7B | 0.19 | 0.16 | 0.10 | 0.03 | 0.09 | 0.01 | 0.10 |
| LightGen, 32 | 1.03 | 1.00 | 3.4B | 0.99 | 0.60 | 0.83 | 0.39 | 0.15 | 0.33 | 0.55 |
| +Step=16 | $0.75_{-0.28}$ | 1.37 | 3.4B | $0.99_{+0.00}$ | $0.59_{-0.01}$ | $0.85_{+0.02}$ | $0.41_{+0.02}$ | $0.15_{+0.00}$ | $0.30_{-0.03}$ | $0.55_{+0.00}$ |
| +LazyMAR | $0.51_{-0.52}$ | 2.00 | 3.4B | $0.99_{+0.00}$ | $0.59_{-0.01}$ | $0.82_{-0.01}$ | $0.40_{+0.01}$ | $0.16_{+0.01}$ | $0.28_{-0.05}$ | $0.54_{-0.01}$ |
| +GtR (Ours) | $0.31_{-0.72}$ | 3.32 | 3.4B | $0.99_{+0.00}$ | $0.58_{-0.02}$ | $0.86_{+0.03}$ | $0.41_{+0.02}$ | $0.14_{-0.01}$ | $0.35_{+0.02}$ | $0.56_{+0.01}$ |
| +Step=12 | $0.68_{-0.35}$ | 1.52 | 3.4B | $0.99_{+0.00}$ | $0.59_{-0.01}$ | $0.88_{+0.05}$ | $0.36_{-0.03}$ | $0.12_{-0.03}$ | $0.25_{-0.08}$ | $0.53_{-0.02}$ |
| +LazyMAR | $0.43_{-0.60}$ | 2.40 | 3.4B | $0.98_{-0.01}$ | $0.56_{-0.04}$ | $0.85_{+0.02}$ | $0.37_{-0.02}$ | $0.13_{-0.02}$ | $0.25_{-0.08}$ | $0.53_{-0.02}$ |
| +GtR (Ours) | $0.27_{-0.76}$ | 3.82 | 3.4B | $1.00_{+0.01}$ | $0.60_{+0.00}$ | $0.84_{+0.01}$ | $0.39_{+0.00}$ | $0.14_{-0.01}$ | $0.35_{+0.02}$ | $0.55_{+0.00}$ |

Table 3: **Ablation studies** for GtR and FTS effectiveness on class-conditional generation. GtR* applies GtR to MAR's encoder-decoder, GtR† applies GtR to MAR's diffusion.

| GtR* | GtR† | FTS | Latency (GPU/s)↓ | Latency (CPU/s)↓ | FLOPs (T)↓ | Speed↑ | FID↓ | IS↑ |
|---|---|---|---|---|---|---|---|---|
| ✗ | ✗ | ✗ | $0.59_{-0.22}$ | $76.05_{-28.61}$ | $45.13_{-19.39}$ | 1.43 | $1.64_{+0.05}$ | $297.3_{-1.8}$ |
| ✓ | ✗ | ✗ | $0.29_{-0.52}$ | $35.27_{-69.39}$ | $22.19_{-42.33}$ | 2.90 | $1.70_{+0.11}$ | $300.1_{+1.0}$ |
| ✗ | ✓ | ✗ | $0.43_{-0.38}$ | $53.58_{-51.08}$ | $33.92_{-30.60}$ | 1.90 | $1.59_{+0.00}$ | $300.4_{+1.3}$ |
| ✓ | ✓ | ✗ | $0.22_{-0.59}$ | $27.02_{-77.64}$ | $17.28_{-47.24}$ | 3.73 | $1.65_{+0.06}$ | $303.4_{+4.3}$ |
| ✓ | ✓ | ✓ | $0.22_{-0.59}$ | $27.93_{-76.73}$ | $17.34_{-47.18}$ | 3.72 | $1.59_{+0.00}$ | $304.4_{+5.3}$ |

Table 4: **Ablation studies** of high-frequency pivot token selection methods in Frequency-Weighted Token Selection (FTS). Evaluation on ImageNet 256×256 during the reconstruction stage.

| Method | FLOPs(T)↓ | Speed↑ | FID↓ | IS↑ |
|---|---|---|---|---|
| Origin | 64.52 | 1.00 | 1.59 | 299.1 |
| **+Random** | $17.34_{-47.18}$ | 3.72 | $1.64_{+0.05}$ | $304.5_{+5.4}$ |
| **+Low-Freq.** | $17.34_{-47.18}$ | 3.72 | $1.64_{+0.05}$ | $301.5_{+2.4}$ |
| **+Full-Enhanced** | $18.04_{-46.48}$ | 3.58 | $1.65_{+0.06}$ | $301.6_{+2.5}$ |
| **+High-Freq. (Ours)** | $17.34_{-47.18}$ | 3.72 | $1.59_{+0.00}$ | $304.4_{+5.3}$ |

Table 5: **Ablation studies** of four token sampling strategies.

| Method | FID↓ | IS↑ |
|---|---|---|
| **Raster** | 24.61 | 120.6 |
| **Subsample** | 5.19 | 247.4 |
| **Random** | 1.82 | 288.8 |
| **GtR (Ours)** | 1.59 | 304.4 |

## 4.4 ABLATION STUDY

**Effectiveness of GtR and FTS.** Table 3 shows the ablation study of the proposed GtR and FTS. It is observed that: (1) When applied individually to either the encoder-decoder or diffusion components of MAR, GtR consistently delivers significant computational gains while maintaining generation quality. (2) When GtR is applied simultaneously to both the encoder-decoder and diffusion, we achieve a 3.73× speedup with only a marginal increase in FID of 0.06 compared to the original MAR. (3) When FTS is further applied, the best results are achieved.

**Token Selection Strategies.** As shown in Table 4, we evaluate four token selection strategies during the reconstruction stage: *Random*: random token selection; *Full-Enhanced*: apply enhanced diffusion steps to all tokens in the reconstruction stage; *High-Freq.*: top 10% tokens with highest importance scores $s^i$; *Low-Freq.*: top 10% tokens with lowest importance scores $s^i$. *Full-Enhanced* and *Low-Freq.* both underperform *Random*, indicating that low-frequency tokens are unsuitable for enhanced diffusion sampling. *High-Freq.* applies enhanced diffusion steps to fine-grained detail tokens, enabling the capture of complex local patterns and textural nuances.

**Impact of Sampling Order.** Table 5 shows the ablation study of different token sampling orders using MAR-H: *Raster* (top-left to bottom-right), *Subsample* (4-quadrant raster), *Random* (permutation), and *GtR* (Generation-then-Reconstruction). *Raster* performs worst because predicted tokens are spatially adjacent. *Subsample* outperforms *Raster* by establishing global structure through quadrant distribution but still suffers from adjacent token prediction within quadrants. *Random* achieves better

performance by mitigating spatial adjacency but may create large blank regions in later generation stages, while *GtR* systematically addresses these limitations for optimal performance.

## 5 CONCLUSION

In this paper, we introduced Generation then Reconstruction (GtR), a training-free hierarchical sampling strategy that significantly accelerates MARs by decomposing generation into structure creation and detail reconstruction stages. By exploiting the observation that spatially adjacent tokens tend to influence each other and that "reconstruction" is considerably easier than "creation", GtR brings significant acceleration without loss of generation quality. We further proposed Frequency-Weighted Token Selection (FTS) to allocate computational resources based on token complexity. Through comprehensive experiments on ImageNet and text-to-image generation, we demonstrated $3.72\times$ acceleration while maintaining comparable quality, establishing a practical framework for efficient parallel visual generation that advances the applicability of MARs in real-world scenarios.

## 6 ACKNOWLEDGEMENT

This research was supported by the Shanghai Science and Technology Program (Grant No. 25ZR1402278) and CCF- Baidu Open Fund.

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

# A APPENDIX

## A.1 USE OF LLMS

During the preparation of this manuscript, we used Large Language Models (LLMs) as auxiliary tools for language editing and presentation enhancement. These tools did not contribute to the research conception, methodology, experimental design, data analysis, or any scientific conclusions presented in this work.

**Language editing.** We used LLMs to assist with grammatical corrections and sentence clarity improvements. LLM suggestions were carefully reviewed, and we made all final decisions regarding text modifications.

**Presentation formatting.** LLMs provided suggestions for table and figure formatting to enhance visual presentation. All scientific content and data remained unchanged.

**Notation consistency.** LLMs occasionally assisted with ensuring consistency in mathematical notation and formatting throughout the manuscript.

We take full responsibility for all content in this manuscript. The research contributions, experimental work, and scientific analysis are entirely our own intellectual work. LLMs served solely as editing assistance tools and had no involvement in the research process or content generation.

## A.2 ABLATION STUDY ON HYPERPARAMETERS AND ROBUSTNESS

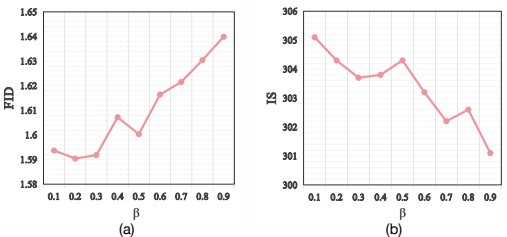 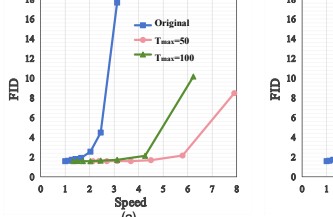 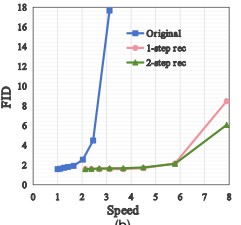

Figure 7: **Sensitivity analysis for FTS percentile $\beta$ on MAR-H at 32 steps on ImageNet 256×256.** (a) FID vs. $\beta$. (b) IS vs. $\beta$. Performance remains stable when $\beta$ ranges from 0.1 to 0.5, but degrades when $\beta$ exceeds 0.5.

Figure 8: **FID vs. Speed trade-off curves on MAR-H at 32 steps on ImageNet 256×256.** (a) Varying initial diffusion steps $T_{\max}$. (b) Varying generation stage steps with reconstruction stage fixed at 1 or 2 steps.

We conduct comprehensive ablation studies to evaluate the sensitivity of GtR to key hyperparameters and demonstrate its robustness across different configurations.

**FTS percentile $\beta$.** Figure 7 presents the sensitivity analysis for the FTS percentile $\beta$ on MAR-H at 32 steps. Performance remains relatively stable when $\beta$ ranges from 0.1 to 0.5, with minimal FID fluctuations. However, when $\beta$ exceeds 0.5, generation quality begins to degrade. This behavior indicates that allocating enhanced diffusion steps to too many tokens becomes suboptimal, as most tokens in the reconstruction stage benefit from strong conditioning information from surrounding generated tokens and do not require additional modeling capacity. Based on these results, we set $\beta = 0.1$ as the default value, ensuring enhanced diffusion steps are allocated only to tokens with complex high-frequency content.

**Trade-off curves for varying configurations.** Figure 8 presents comprehensive trade-off analyses. Figure 8(a) shows the FID vs. Speed curves when varying initial diffusion steps $T_{\max}$. GtR outperforms the original MAR for both $T_{\max} = 50$ and $T_{\max} = 100$ settings across all speedup ratios. Figure 8(b) presents the trade-off when varying generation stage steps while fixing the reconstruction stage to either 1 or 2 steps. Regardless of the reconstruction stage configuration, GtR consistently outperforms the original MAR. Notably, when the speedup ratio exceeds 3.2×, the original MAR exhibits severe quality degradation, whereas GtR maintains stable generation quality even at 8× speedup, validating the robustness of our two-stage design.

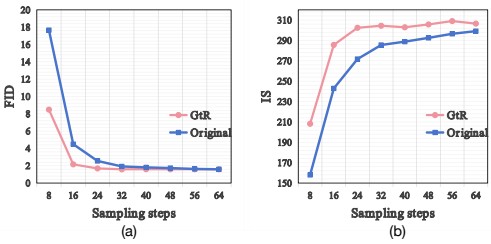

Figure 9: **Impact of sampling steps on MAR generation quality on ImageNet 256×256.** (a) FID vs. Sampling Steps. (b) IS vs. Sampling Steps. GtR consistently achieves superior generation quality compared to original MAR across all sampling steps.

**Impact of sampling steps.** Figure 9 shows the FID and IS curves as a function of sampling steps for both the original MAR and GtR on ImageNet $256 \times 256$. GtR consistently achieves superior generation quality compared to the original MAR across all sampling steps, with the advantage becoming more pronounced as the number of steps decreases. At very low step counts where the original MAR exhibits severe quality degradation, GtR maintains stable performance, demonstrating the effectiveness of our checkerboard partition strategy in preserving generation quality under extreme acceleration settings.

Table 6: **Ablation study on checkerboard pattern selection for MAR-H at 32 steps on ImageNet 256×256.**

| Pattern | FID↓ | IS↑ |
|---|---|---|
| $(i + j) \bmod 2 = 0$ | 1.59 | 304.4 |
| $(i + j) \bmod 2 = 1$ | 1.60 | 302.8 |

**Checkerboard pattern selection.** We evaluate the impact of checkerboard pattern selection by comparing the two complementary partitions: $(i + j) \bmod 2 = 0$ versus $(i + j) \bmod 2 = 1$. As shown in Table 6, both patterns yield comparable generation quality on MAR-H at 32 steps, with marginal differences in FID (1.59 vs 1.60) and IS (304.4 vs 302.8). This validates the robustness of GtR's checkerboard reconstruction strategy regardless of which specific pattern is selected, as both patterns maintain equivalent spatial separation properties that are critical for minimizing token dependencies during generation.

These ablation studies demonstrate that GtR maintains stable performance across reasonable hyper-parameter ranges, enabling straightforward application to different MAR models without extensive hyperparameter tuning.

