# OpenReview forum: "Generation then Reconstruction: Accelerating Masked Autoregressive Models via Two-Stage Sampling"
_ICLR.cc/2026/Conference — ICLR 2026 Poster_

### Official Review · Reviewer_ZCm1 · 2025-10-30

**Soundness:** 3
**Presentation:** 3
**Contribution:** 3
**Rating:** 6
**Confidence:** 4

**Summary:**

Based on the two observations: (1) the spatially adjacent tokens tend to influence each other; and (2) covering more spatial locations indicates creating more information, this paper proposes Generation-then-Reconstruction (GtR), a novel sampling strategy to accelerate image generation in Masked Autoregressive (MAR) models, which decomposes generation into two stages: structure generation establishing global semantic scaffolding, followed by detail reconstruction efficiently completing remaining tokens. It also proposes Frequency Weighted Token Selection (FTS) to offer more computation budget to tokens on image details. Extensive experiments on ImageNet class-conditional generation and text-to-image synthesis demonstrate that GtR achieves significant speedups while maintaining competitive or superior image quality.

**Strengths:**

1.	The "generation-then-reconstruction" paradigm is conceptually novel, which is a training-free hierarchical sampling strategy and expands the mainstream random/sequential generation mode.
2.	The proposed Frequency-Weighted Token Selection (FTS), a training-free strategy that could allocate more diffusion steps to the tokens with more details.
3.	Compared with other accelerated model, the proposed method achieves faster inference speedup for MAR while maintaining comparable quality, which better balance between generation speed and generation quality.

**Weaknesses:**

1.	Some of the organization and arguments in the paper are problematic: a). In line 013, the MAR is the Masked Autoregressive (MAR) models from [1] , but it is not clear whether AR specifically refers to a discrete autoregressive model or a continuous autoregressive model. If it is discrete autoregressive model, the inference efficiency of MAR is lower than AR; b). In line 134, the xAR should not be classified as Next-token autoregressive visual generation, because more than one token is generated at each step.
2.	Although the specific division stages and the average generation rates for each stage are described in section 4.1, the number of inference steps of each stage is unclear. And it is not clear why it is set up like this.
3.	The generated tokens in the generation stage satisfied the condition (i+j) mod 2 = 0 in all experiments. Lack the ablation study about setting the condition (i+j) mod 2 = 1. In addition, lack the ablation study about β.
4.	Inconsistent writing format in section 3.1. “Image Generation as Next-Token Prediction.” and “Image generation as next-set prediction” .

[1] Autoregressive image generation without vector quantization.

**Questions:**

1.	The paper uses K=3 for MAR and K=4 for LightGen. Is this choice empirical or theoretically motivated? How does K affect the quality-speed trade-off?

---

> ### Author Response · Authors · 2025-11-26
> **Response to reviewer  ZCm1 (1/N)**
>
> > **Q4.1.** Some of the organization and arguments in the paper are problematic: a). In line 013, the MAR is the Masked Autoregressive (MAR) models from [1] , but it is not clear whether AR specifically refers to a discrete autoregressive model or a continuous autoregressive model. If it is discrete autoregressive model, the inference efficiency of MAR is lower than AR; b). In line 134, the xAR should not be classified as Next-token autoregressive visual generation, because more than one token is generated at each step.
>
> Thank you for pointing this out. In the abstract, "AR" specifically refers to continuous autoregressive image generation models, rather than discrete token autoregressive models; we will make this definition explicit in the revised version. For xAR, we agree that it should not be categorized as a next-token autoregressive visual generator, since multiple tokens are produced at each step. We will revise the taxonomy in the introduction accordingly.

---

> ### Author Response · Authors · 2025-11-26
> **Response to reviewer  ZCm1 (2/N)**
>
> > **Q4.2.** Although the specific division stages and the average generation rates for each stage are described in section 4.1, the number of inference steps of each stage is unclear. And it is not clear why it is set up like this.
>
> Thank you for raising this concern. We provide a comprehensive explanation of how the inference steps are allocated across stages. Please refer to Appendix A.1 (Inference Steps Allocation Details) for the formal definition and derivation of step allocation.
>
> **We provide a more intuitive understanding:** during the generation stage, half of the total tokens are generated over $T_{\text{total}} - 2$ steps, with the masking ratio decaying from 1 to 0 following the schedule $\left(1 - \left(\frac{t}{T_{\text{total}} - 2}\right)^\alpha\right)$. Subsequently, the reconstruction stage generates the remaining half of the tokens in 2 steps. This allocation strategy ensures progressively increasing parallelism ratio as generation proceeds, enabling efficient acceleration. It is important to note that the number of sub-stages within the generation stage only affects the ordering of token generation (i.e., which tokens are generated first) rather than the parallelism ratio (i.e., how many tokens are generated per step).
>
> **Importantly, we did not perform extensive hyperparameter tuning:** key parameters such as $\alpha=2.7$ and the 2-step reconstruction stage remain consistent across all models. The choice of the number of stages $K$ depends primarily on the total token count $N$, and the response to Q4.5 demonstrate strong robustness to variations in $K$.
>
> **To further validate GtR's generalizability and robustness,** we conducted experiments on Harmon, a more advanced MAR-based text-to-image model. As shown in Rebuttal Table 4.1 below, GtR achieves substantial acceleration while improving generation quality across multiple settings. Specifically, when using 16 steps with GtR, we achieve a 4.71× speedup (3.72s → 0.79s) compared to Harmon with 64 steps, while simultaneously improving the GenEval Overall score from 0.82 to 0.83. Under extreme settings (8 steps), GtR substantially outperforms the baseline (Overall score of 0.81 vs 0.74), further validating the effectiveness and robustness of GtR.
>
> **Rebuttal Table 4.1: GenEval performance comparison on Harmon at 512×512 resolution**
>
> | Method           | Latency(GPU/s)↓ | Speed↑ | Single Obj.↑ | Two Obj.↑ | Colors↑ | Counting↑ | Position↑ | Color Attri.↑ | Overall↑ |
> |------------------|----------------:|-------:|-------------:|----------:|--------:|----------:|----------:|--------------:|---------:|
> | steps=64         |            3.72 |   1.00 |         1.00 |      0.92 |    0.87 |      0.78 |      0.67 |          0.70 |     0.82 |
> | steps=32         |            2.17 |   1.71 |         1.00 |      0.93 |    0.88 |      0.76 |      0.64 |          0.70 |     0.82 |
> | **&nbsp;&nbsp;+GtR** |        **1.14** | **3.26** |     **0.99** |  **0.92** | **0.88** |  **0.80** |  **0.68** |      **0.73** | **0.83** |
> | steps=16         |            1.42 |   2.62 |         0.99 |      0.90 |    0.87 |      0.73 |      0.64 |          0.71 |     0.81 |
> | **&nbsp;&nbsp;+GtR** |        **0.79** | **4.71** |     **1.00** |  **0.92** | **0.88** |  **0.79** |  **0.67** |      **0.73** | **0.83** |
> | steps=8          |            1.06 |   3.51 |         0.99 |      0.85 |    0.84 |      0.60 |      0.60 |          0.57 |     0.74 |
> | **&nbsp;&nbsp;+GtR** |        **0.61** | **6.10** |     **0.99** |  **0.89** | **0.90** |  **0.74** |  **0.66** |      **0.69** | **0.81** |
>
> The specific configurations for MAR and LightGen described in Section 4.1 are shown in Rebuttal Tables 4.2 and 4.3.
>
> **Rebuttal Table 4.2: MAR configuration ($N=256$, $T_{\text{total}}=32$, $K=3$ stages)**
>
> | Stage                | Steps | Tokens Generated | Generation Rate (tokens/step) |
> |----------------------|------:|-----------------:|------------------------------:|
> | Generation Stage 1   |    24 |               64 |                          2.67 |
> | Generation Stage 2   |     6 |               64 |                         10.67 |
> | Reconstruction Stage |     2 |              128 |                         64.00 |
>
> **Rebuttal Table 4.3: LightGen configuration ($N=1024$, $T_{\text{total}}=16$, $K=4$ stages)**
>
> | Stage                | Steps | Tokens Generated | Generation Rate (tokens/step) |
> |----------------------|------:|-----------------:|------------------------------:|
> | Generation Stage 1   |     8 |              128 |                         16.00 |
> | Generation Stage 2   |     3 |              128 |                         42.67 |
> | Generation Stage 3   |     3 |              256 |                         85.33 |
> | Reconstruction Stage |     2 |              512 |                        256.00 |
>
> These configurations demonstrate how GtR adaptively allocates steps across different model scales while maintaining consistent hyperparameters, showcasing its strong generalizability.

---

> ### Author Response · Authors · 2025-11-26
> **Response to reviewer  ZCm1 (3/N)**
>
> > **Q4.3.** The generated tokens in the generation stage satisfied the condition (i+j) mod 2 = 0 in all experiments. Lack the ablation study about setting the condition (i+j) mod 2 = 1. In addition, lack the ablation study about β.
>
> Thank you for this suggestion. **We have conducted ablation studies on the checkerboard pattern selection** with MAR-H at $T=32$ steps as presented in Rebuttal Table 4.4 below. Results show that $(i+j) \bmod 2 = 0$ achieves slightly better performance (FID: 1.59, IS: 304.4) compared to $(i+j) \bmod 2 = 1$ (FID: 1.60, IS: 302.8). The marginal difference (∆FID=0.01, ∆IS=1.6) indicates that both patterns yield comparable generation quality, validating the robustness of GtR's checkerboard reconstruction strategy regardless of pattern selection. We thank you again for your valuable feedback and will include these results in the Appendix of the revised paper.
>
> **Rebuttal Table 4.4: Ablation study on checkerboard pattern selection**
>
> | Pattern                 | FID↓ |   IS↑ |
> |-------------------------|-----:|------:|
> | $(i+j) \bmod 2 = 0$     | 1.59 | 304.4 |
> | $(i+j) \bmod 2 = 1$     | 1.60 | 302.8 |
>
>
> **Regarding the ablation study on $\beta$ (FTS percentile):** We evaluate the FTS percentile $\beta$ on MAR-H at 32 steps as presented in Rebuttal Table 4.5. Performance remains stable when $\beta$ ranges from 0.1 to 0.5. However, when $\beta$ exceeds 0.5, generation quality degrades. This suggests that allocating enhanced diffusion steps to too many tokens is suboptimal, as most tokens in the reconstruction stage are easier to generate due to strong conditioning from surrounding tokens. We set $\beta=0.1$ as the default value, ensuring enhanced diffusion steps are allocated only to tokens with complex high-frequency content.
>
> **Rebuttal Table 4.5: Ablation study on FTS percentile $\beta$ for MAR-H at 32 steps on ImageNet 256×256**
>
> | $\beta$ | FID↓ |   IS↑ |
> |--------:|-----:|------:|
> |     0.1 | 1.59 | 304.4 |
> |     0.2 | 1.59 | 304.3 |
> |     0.3 | 1.59 | 303.7 |
> |     0.4 | 1.60 | 303.8 |
> |     0.5 | 1.60 | 304.3 |
> |     0.6 | 1.61 | 303.2 |
> |     0.7 | 1.62 | 302.2 |
> |     0.8 | 1.63 | 302.6 |
> |     0.9 | 1.64 | 301.1 |
>
> We thank you for pointing out this missing ablation and will include these detailed results in the Appendix of the revised paper.
>
> > **Q4.4.** Inconsistent writing format in section 3.1. "Image Generation as Next-Token Prediction." and "Image generation as next-set prediction" .
>
> Thank you for pointing this out. We will fix the formatting inconsistency in the revised paper.

---

> ### Author Response · Authors · 2025-11-26
> **Response to reviewer  ZCm1 (4/N)**
>
> > **Q4.5.** The paper uses K=3 for MAR and K=4 for LightGen. Is this choice empirical or theoretically motivated? How does K affect the quality-speed trade-off?
>
> Thank you for this question. The choice of $K$ is primarily determined by the total number of tokens $N$: images with more tokens use more stages to enable establishing the fundamental semantic structure in the early stages as soon as possible. It is important to note that varying $K$ only changes the order in which tokens are generated, but does not affect the total computational cost, as explained in Q4.2.
>
> We conduct ablation studies to evaluate sensitivity to $K$. As shown in Rebuttal Tables 4.6 and 4.7, performance remains stable across different $K$ values. On MAR, varying $K$ from 2 to 4 yields minimal FID fluctuations (1.59-1.64). On LightGen, the Overall score remains at 0.55-0.56. These results demonstrate that GtR is robust to $K$, with quality primarily determined by total steps.
>
> **Rebuttal Table 4.6: Ablation study on $K$ for MAR-H at 32 steps on ImageNet 256×256**
>
> | $K$ | FID↓ |   IS↑ |
> |----:|-----:|------:|
> |   2 | 1.64 | 304.1 |
> |   3 | 1.59 | 304.4 |
> |   4 | 1.63 | 304.2 |
>
> **Rebuttal Table 4.7: Ablation study on $K$ for LightGen at 16 steps on GenEval**
>
> | $K$ | Single Obj.↑ | Two Obj.↑ | Colors↑ | Counting↑ | Position↑ | Color Attri.↑ | Overall↑ |
> |----:|-------------:|----------:|--------:|----------:|----------:|--------------:|---------:|
> |   2 |         0.99 |      0.59 |    0.84 |      0.43 |      0.13 |          0.34 |     0.56 |
> |   3 |         0.99 |      0.61 |    0.84 |      0.38 |      0.13 |          0.37 |     0.55 |
> |   4 |         0.99 |      0.58 |    0.86 |      0.41 |      0.14 |          0.35 |     0.56 |
>
> In our paper, for MAR with $N=256$ tokens, we set $K=3$ with stage allocations of 64, 64, and 128 tokens. For LightGen with $N=1024$ tokens, we set $K=4$ with allocations of 128, 128, 256, and 512 tokens.
>
> As shown in Q4.2, we further validated GtR on Harmon, a more advanced MAR-based model, maintaining the same $K$ and other hyperparameters ($\alpha$, reconstruction stage steps) as LightGen. Additionally, we validated these hyperparameters in challenging scenarios including detail-dense images with weak global structure (Rebuttal Table 4.8) and non-square layouts at 512×384 resolution (Rebuttal Table 4.9). GtR achieves 4.71× speedup while maintaining comparable generation quality in both scenarios, further demonstrating the robustness of our hyperparameter choices across diverse content types and aspect ratios.
>
> **Rebuttal Table 4.8: Image Reward comparison on Harmon with detail-dense prompts and weak structure**
>
> | Method           | Latency(GPU/s)↓ | Speed↑ | Image Reward↑ |
> |------------------|----------------:|-------:|--------------:|
> | steps=64         |            3.72 |   1.00 |          0.90 |
> | steps=32         |            2.17 |   1.71 |          0.90 |
> | **&nbsp;&nbsp;+GtR** |        **1.14** | **3.26** |      **0.90** |
> | steps=16         |            1.42 |   2.62 |          0.86 |
> | **&nbsp;&nbsp;+GtR** |        **0.79** | **4.71** |      **0.90** |
> | steps=8          |            1.06 |   3.51 |          0.69 |
> | **&nbsp;&nbsp;+GtR** |        **0.61** | **6.10** |      **0.79** |
>
> **Rebuttal Table 4.9: Image Reward comparison on Harmon with non-square layouts (512×384)**
>
> | Method           | Latency(GPU/s)↓ | Speed↑ | Image Reward↑ |
> |------------------|----------------:|-------:|--------------:|
> | steps=64         |            3.72 |   1.00 |          0.82 |
> | steps=32         |            2.17 |   1.71 |          0.78 |
> | **&nbsp;&nbsp;+GtR** |        **1.14** | **3.26** |      **0.83** |
> | steps=16         |            1.42 |   2.62 |          0.76 |
> | **&nbsp;&nbsp;+GtR** |        **0.79** | **4.71** |      **0.82** |
> | steps=8          |            1.06 |   3.51 |          0.55 |
> | **&nbsp;&nbsp;+GtR** |        **0.61** | **6.10** |      **0.67** |

---

### Official Review · Reviewer_FhEK · 2025-10-30

**Soundness:** 3
**Presentation:** 3
**Contribution:** 3
**Rating:** 8
**Confidence:** 3

**Summary:**

This paper proposes Generation-then-Reconstruction (GtR), a training-free, two-stage sampling strategy to accelerate Masked Autoregressive (MAR) visual generation. GtR first generates a spatially non-adjacent “skeleton” of tokens (checkerboard / conservative generation stage) to establish global semantic structure, then reconstructs the remaining tokens in very few highly-parallel steps. The authors also introduce Frequency-Weighted Token Selection (FTS) to allocate extra diffusion steps to tokens with high high-frequency energy (i.e., fine details). On ImageNet class-conditional and text-to-image setups the paper reports ~3.72× speedup for MAR-H with comparable FID/IS. The method is presented as training-free and applicable to existing MAR pipelines.

**Strengths:**

- **Simple, practical idea with strong engineering value.** Splitting generation into “creation” (structure) and “reconstruction” (detail) is intuitive and requires no model retraining, which makes it immediately usable in deployed MAR systems.
- **Clear empirical speed/quality benefit.** The reported ≈3.72× acceleration on MAR-H while maintaining generation quality is compelling evidence of practical value.
- **Ablations and comparisons.** The paper contains ablations across sampling orders and token selection strategies (Raster/Subsample/Random/GtR and High-Freq/Random/Full-Enhanced), which strengthens the empirical claims.

**Weaknesses:**

- **Limited analysis of failure modes and generality.** The paper demonstrates gains on ImageNet and a text→image generator, but it is unclear how GtR behaves with different tokenizers (VQ variants, continuous latents), non-square layouts, or with very detail-dense images where global structure is weak. The paper makes the empirical claim that reconstruction is “much easier than creation” but does not deeply analyze where this breaks down.
- **Limited theoretical justification for checkerboard schedule.** The checkerboard / stage partition is intuitively motivated (reduce adjacent simultaneous predictions) but lacks theoretical analysis or simple metrics demonstrating why it is near-optimal vs. other structured partitions. The ablations show GtR > Random, but more insight would help adoption.

**Questions:**

- The paper report “3.72× speedup”, but it is not clear whether this is measured as (a) total wall-clock on standard hardware, (b) encoder FLOPs only, (c) decoder/diffusion FLOPs, or (d) a mix. A breakdown (encoder vs diffusion vs IO) and experiments on multiple hardware targets would increase reproducibility and confidence.
- I recommend conducting a Hyperparameter sensitivity study, including experiments that sweep K, Trec, β (FTS percentile), and Tdetail to show how sensitive quality and speed are. A small table/heatmap showing FID vs speed for several settings would help practitioners choose defaults.

---

> ### Author Response · Authors · 2025-11-26
> **Response to reviewer  FhEK (1/N)**
>
> > **Q3.1.** Limited analysis of failure modes and generality. The paper demonstrates gains on ImageNet and a text→image generator, but it is unclear how GtR behaves with different tokenizers (VQ variants, continuous latents), non-square layouts, or with very detail-dense images where global structure is weak. The paper makes the empirical claim that reconstruction is "much easier than creation" but does not deeply analyze where this breaks down.
>
> Thank you for this feedback. We conduct additional experiments to validate GtR's generalization across the challenging scenarios you mentioned and analyze potential failure modes.
>
> **Detail-dense images with weak global structure.** We collected 80 prompts describing detail-dense images with weak global structure and evaluated GtR on Harmon. As shown in Rebuttal Table 3.1 below, at 16 steps GtR achieves a 4.71× speedup compared to Harmon with 64 steps while maintaining comparable generation quality (Image Reward: 0.90 vs 0.90). At 8 steps, GtR improves Image Reward by 0.10 compared to Harmon with 8 steps (from 0.69 to 0.79), demonstrating that GtR remains effective even under extreme settings.
>
> **Rebuttal Table 3.1: Image Reward comparison on Harmon with detail-dense prompts and weak structure**
>
> | Method           | Latency(GPU/s)↓ | Speed↑ | Image Reward↑ |
> |------------------|----------------:|-------:|--------------:|
> | steps=64         |            3.72 |   1.00 |          0.90 |
> | steps=32         |            2.17 |   1.71 |          0.90 |
> | **&nbsp;&nbsp;+GtR** |        **1.14** | **3.26** |      **0.90** |
> | steps=16         |            1.42 |   2.62 |          0.86 |
> | **&nbsp;&nbsp;+GtR** |        **0.79** | **4.71** |      **0.90** |
> | steps=8          |            1.06 |   3.51 |          0.69 |
> | **&nbsp;&nbsp;+GtR** |        **0.61** | **6.10** |      **0.79** |
>
> **Non-square layouts.** We evaluated GtR on 100 prompts at 512×384 resolution on Harmon. As shown in Rebuttal Table 3.2 below, at 16 steps GtR achieves a 4.71× speedup compared to Harmon with 64 steps while maintaining comparable generation quality (Image Reward: 0.82 vs original 0.82).
>
> **Rebuttal Table 3.2: Image Reward comparison on Harmon with non-square layouts (512×384)**
>
> | Method           | Latency(GPU/s)↓ | Speed↑ | Image Reward↑ |
> |------------------|----------------:|-------:|--------------:|
> | steps=64         |            3.72 |   1.00 |          0.82 |
> | steps=32         |            2.17 |   1.71 |          0.78 |
> | **&nbsp;&nbsp;+GtR** |        **1.14** | **3.26** |      **0.83** |
> | steps=16         |            1.42 |   2.62 |          0.76 |
> | **&nbsp;&nbsp;+GtR** |        **0.79** | **4.71** |      **0.82** |
> | steps=8          |            1.06 |   3.51 |          0.55 |
> | **&nbsp;&nbsp;+GtR** |        **0.61** | **6.10** |      **0.67** |
>
> **Regarding why reconstruction is "much easier than creation."** The checkerboard partition ensures that when the generation stage finishes, every ungenerated token is surrounded by generated tokens. As a result, each ungenerated token receives strong conditioning information from its neighbors, which significantly reduces the dependencies among ungenerated tokens. Concretely, the joint distribution of ungenerated tokens can be accurately approximated as a product of independent distributions, making the reconstruction task much easier than the original generation task where tokens have complex interdependencies.
>
> However, GtR's performance is bounded by the underlying model capacity. For cases where the baseline model produces poor results, GtR faces similar challenges. We will investigate these challenging failure modes in future work to further improve GtR's robustness. We thank you again for your valuable feedback.

---

> ### Author Response · Authors · 2025-11-26
> **Response to reviewer  FhEK (2/N)**
>
> > **Q3.2.** Limited theoretical justification for checkerboard schedule. The checkerboard / stage partition is intuitively motivated (reduce adjacent simultaneous predictions) but lacks theoretical analysis or simple metrics demonstrating why it is near-optimal vs. other structured partitions. The ablations show GtR > Random, but more insight would help adoption.
>
> Thank you for this insightful question. We provide a comprehensive theoretical analysis grounded in information theory in Appendix A.4 to justify why the checkerboard partition is near-optimal for MAR sampling.
>
> The theoretical foundation lies in minimizing the approximation error inherent in MAR's sampling process. We formulate the optimization as a two-stage problem: (1) In the generation stage, minimizing intra-set Mutual Information MI(S_G) by maximizing spatial separation among tokens, which corresponds to the Maximum Independent Set (MIS) problem on the grid graph. Since the grid graph is bipartite, the checkerboard partition uniquely constitutes the MIS, optimally eliminating dominant dependencies. (2) In the reconstruction stage, minimizing conditional entropy H(S_R|S_G) by exploiting the local Markov property. The checkerboard partition ensures that for every ungenerated token, its entire Markov Blanket is observed in the generation stage, providing the strongest possible local conditioning and enabling rapid parallel reconstruction.
>
> **This theoretical framework explains why GtR outperforms random sampling:** random sampling lacks structural guarantees for spatial separation during generation and cannot ensure complete Markov Blanket observation during reconstruction, leading to higher approximation errors. Please refer to Appendix A.4 for the complete theoretical derivation and formal proofs.

---

> ### Author Response · Authors · 2025-11-26
> **Response to reviewer  FhEK (3/N)**
>
> > **Q3.3.** The paper report "3.72× speedup", but it is not clear whether this is measured as (a) total wall-clock on standard hardware, (b) encoder FLOPs only, (c) decoder/diffusion FLOPs, or (d) a mix. A breakdown (encoder vs diffusion vs IO) and experiments on multiple hardware targets would increase reproducibility and confidence.
>
> Thank you for this careful review. The reported 3.72× speedup refers to total wall-clock time measured on standard hardware (RTX 4090 GPU with Intel 12900K CPU). To enhance reproducibility, we present detailed latency breakdowns across encoder-decoder, diffusion, and IO components on two hardware platforms in Rebuttal Tables 3.3 and 3.4.
>
> As shown in Rebuttal Table 3.3 (RTX 4090), at 32 steps GtR achieves a 3.61× speedup compared to MAR-H with 64 steps while maintaining identical generation quality (FID: 1.59 vs original 1.59). Notably, GtR enables efficient encoder-decoder processing (0.19s vs original 0.61s) and substantially reduces diffusion latency (0.04s vs original 0.21s). At the extreme setting of 16 steps, GtR achieves a 5.93× speedup relative to the 64 step baseline while substantially improving quality over MAR-H with 16 steps (FID: 2.16 vs original 4.49, IS: 285.6 vs original 242.9), with encoder-decoder latency further reduced to 0.11s (vs original 0.16s). On A100 (Rebuttal Table 3.4), consistent patterns are observed with 4.17× speedup at 32 steps and 7.14× at 16 steps, validating hardware-agnostic efficiency gains.
>
> **Rebuttal Table 3.3: Detailed latency breakdown on ImageNet 256×256 (RTX 4090)**
>
> | Method           | Latency(GPU/s)↓ | Speed↑ | Latency(Enc-Dec)↓ | Latency(Diffusion)↓ | Latency(IO)↓ | FID↓ |   IS↑ |
> |------------------|----------------:|-------:|------------------:|--------------------:|-------------:|-----:|------:|
> | MAR-H, 64        |            0.83 |   1.00 |              0.61 |                0.21 |         0.01 | 1.59 | 299.1 |
> | MAR-H, 32        |            0.49 |   1.69 |              0.31 |                0.18 |         0.01 | 1.92 | 285.5 |
> | **&nbsp;&nbsp;+GtR** |        **0.23** | **3.61** |          **0.19** |            **0.04** |     **0.01** | **1.59** | **304.4** |
> | MAR-H, 16        |            0.34 |   2.44 |              0.16 |                0.17 |         0.01 | 4.49 | 242.9 |
> | **&nbsp;&nbsp;+GtR** |        **0.14** | **5.93** |          **0.10** |            **0.04** |     **0.01** | **2.16** | **285.6** |
>
> **Rebuttal Table 3.4: Detailed latency breakdown on ImageNet 256×256 (A100)**
>
> | Method           | Latency(GPU/s)↓ | Speed↑ | Latency(Enc-Dec)↓ | Latency(Diffusion)↓ | Latency(IO)↓ | FID↓ |   IS↑ |
> |------------------|----------------:|-------:|------------------:|--------------------:|-------------:|-----:|------:|
> | MAR-H, 64        |            1.00 |   1.00 |              0.53 |                0.47 |         0.01 | 1.59 | 299.1 |
> | MAR-H, 32        |            0.54 |   1.85 |              0.27 |                0.27 |         0.01 | 1.92 | 285.5 |
> | **&nbsp;&nbsp;+GtR** |        **0.24** | **4.17** |          **0.16** |            **0.08** |     **0.01** | **1.59** | **304.4** |
> | MAR-H, 16        |            0.34 |   2.94 |              0.14 |                0.20 |         0.01 | 4.49 | 242.9 |
> | **&nbsp;&nbsp;+GtR** |        **0.14** | **7.14** |          **0.08** |            **0.05** |     **0.01** | **2.16** | **285.6** |

---

> ### Author Response · Authors · 2025-11-26
> **Response to reviewer  FhEK (4/N)**
>
> > **Q3.4.** I recommend conducting a Hyperparameter sensitivity study, including experiments that sweep K, Trec, β (FTS percentile), and Tdetail to show how sensitive quality and speed are. A small table/heatmap showing FID vs speed for several settings would help practitioners choose defaults.
>
> Thank you for this suggestion. We conduct systematic sensitivity analyses across key hyperparameters to validate the robustness of our method and provide practitioners with guidance for parameter selection.
>
> **We first examine the sensitivity to the number of stages $K$.** As shown in Rebuttal Tables 3.5 and 3.6 below, varying $K$ from 2 to 4 yields minimal performance fluctuations on both MAR and LightGen. On MAR at 32 steps, FID remains stable between 1.59 and 1.64 while IS consistently stays around 304. Similarly, on LightGen at 16 steps, the GenEval Overall score fluctuates only within 0.55 to 0.56 across different $K$ values, demonstrating that our framework exhibits robustness to this hyperparameter.
>
> **Rebuttal Table 3.5: Ablation study on $K$ for MAR-H at 32 steps on ImageNet 256×256**
>
> | $K$ | FID↓ |   IS↑ |
> |----:|-----:|------:|
> |   2 | 1.64 | 304.1 |
> |   3 | 1.59 | 304.4 |
> |   4 | 1.63 | 304.2 |
>
> **Rebuttal Table 3.6: Ablation study on $K$ for LightGen at 16 steps on GenEval**
>
> | $K$ | Single Obj.↑ | Two Obj.↑ | Colors↑ | Counting↑ | Position↑ | Color Attri.↑ | Overall↑ |
> |----:|-------------:|----------:|--------:|----------:|----------:|--------------:|---------:|
> |   2 |         0.99 |      0.59 |    0.84 |      0.43 |      0.13 |          0.34 |     0.56 |
> |   3 |         0.99 |      0.61 |    0.84 |      0.38 |      0.13 |          0.37 |     0.55 |
> |   4 |         0.99 |      0.58 |    0.86 |      0.41 |      0.14 |          0.35 |     0.56 |
>
> **For the FTS percentile $\beta$**, as shown in Figure 7 in our Appendix, performance remains relatively stable when $\beta$ ranges from 0.1 to 0.5. However, when $\beta$ exceeds 0.5, generation quality begins to degrade, suggesting that allocating excessive diffusion steps to too many tokens is suboptimal for modeling in the later reconstruction stage.
>
> **Regarding the detailed diffusion steps $T_{\text{detail}}$**, we evaluate its impact on MAR-H at 32 steps. As shown in Rebuttal Table 3.7, setting $T_{\text{detail}}=50$ achieves the best performance with FID of 1.59 and IS of 304.4. When $T_{\text{detail}}$ is set too low (20 or 40), the allocated diffusion steps are insufficient for accurately modeling complex texture regions, resulting in slightly degraded quality. Conversely, excessively high values (60 or 80) also fail to improve performance. This suggests that in the later reconstruction stage, high-frequency tokens benefit from strong conditioning information accumulated from surrounding generated tokens, and excessive diffusion steps may become suboptimal.
>
> **Rebuttal Table 3.7: Sensitivity to $T_{\text{detail}}$ on MAR-H (32 steps, ImageNet 256×256)**
>
> | $T_{\text{detail}}$ | FID↓ |   IS↑ |
> |--------------------:|-----:|------:|
> |                  20 | 1.65 | 303.4 |
> |                  40 | 1.64 | 302.9 |
> |                  50 | 1.59 | 304.4 |
> |                  60 | 1.61 | 303.9 |
> |                  80 | 1.63 | 300.0 |
>
> These ablation studies demonstrate that GtR maintains stable performance across reasonable hyperparameter ranges, with our default settings ($K=3$ for MAR, $K=4$ for LightGen, $\beta=0.1$, $T_{\text{detail}}=50$) achieving near-optimal results. This robustness enables straightforward application to different MAR models without extensive hyperparameter tuning.

---

### Official Review · Reviewer_fAqu · 2025-11-01

**Soundness:** 3
**Presentation:** 2
**Contribution:** 2
**Rating:** 4
**Confidence:** 4

**Summary:**

This paper proposed a two-stage sampling strategy (Generation then Reconstruction, GtR) for MAR. The goal is to sample images with fewer autoregressive steps while maintaining comparable quality. Specifically, the generation stage samples image tokens in a checkerboard sequence, using more steps to establish global semantic scaffolding and structure. The reconstruction stage then samples the remaining tokens with fewer steps to fill in the details. Additionally, the paper introduces a Frequency-Weighted Token Selection strategy for choosing diffusion steps. Experimental results demonstrate that this GtR strategy can accelerate sampling speed.

**Strengths:**

1. The proposed sampling strategy is effective and intuitively easy to understand.

2. The paper is clearly written.

3. The performance in ImageNet-256x256 is impressive.

**Weaknesses:**

1. Limited Generalizability: The method's generalizability is a significant concern. While it performs well on ImageNet, its performance drops considerably on the T2I task (as shown in Table 2). This suggests that the proposed sampling strategy might be an overfitted, heavily-tuned solution for the ImageNet dataset and may not generalize well to other datasets or tasks. Moreover, the proposed strategy's applicability seems limited, as it is designed specifically for the MAR.

2. The novelty of the work is limited. The core concept of establishing a global structure before refining details ("global-to-detail") has been explored in prior work like MUSE[1] and Hi-MAR[2] (missing reference and comparison). The paper's main contribution appears to be adapting this concept specifically to the sampling process of MAR.

3. Insufficient Experiments: The paper only reports results for a very sparse set of sampling steps (e.g., steps 8 and 16 for C2I; steps 12 and 16 for T2I). It is unclear why step 8 was omitted for the T2I task, and the performance at other step counts remains unknown. The authors should provide the curves (e.g., FID vs. Sampling Steps) to show the full performance trend. Moreover, there are no trade-off curves to analyze the FID vs. speed when: (a) varying the step of the generation and reconstruction stages, (b) varying the autoregressive steps and the diffusion steps.

4. Do not provide an introduction to the baseline methods (Halton, DiSA, LazyMAR).

I am open to raising my score if the authors can address these concerns in their rebuttal.

[1]Chang, Huiwen, et al. "Muse: Text-to-image generation via masked generative transformers." ICML 2023.

[2]Zheng, Guangting, et al. "Hierarchical Masked Autoregressive Models with Low-Resolution Token Pivots." ICML 2025.

**Questions:**

Please refer to weaknesses.

---

> ### Author Response · Authors · 2025-11-26
> **Response to reviewer  fAqu (1/N)**
>
> > **Q2.1.** Limited Generalizability: The method's generalizability is a significant concern. While it performs well on ImageNet, its performance drops considerably on the T2I task (as shown in Table 2). This suggests that the proposed sampling strategy might be an overfitted, heavily-tuned solution for the ImageNet dataset and may not generalize well to other datasets or tasks. Moreover, the proposed strategy's applicability seems limited, as it is designed specifically for the MAR.
>
> Thank you for this valuable feedback. We appreciate the opportunity to clarify GtR's performance on text-to-image generation tasks. As shown in Table 2 of the revised paper, when applied to LightGen with 12 steps, GtR achieves a 3.82× speedup (1.03s → 0.27s) while maintaining the same GenEval score of 0.55 as the original LightGen.
>
> To further demonstrate GtR's generalizability, we conducted additional experiments on Harmon, a more advanced MAR-based text-to-image model. As presented in Rebuttal Table 2.1 below, GtR achieves substantial acceleration while improving generation quality across multiple settings. Specifically, when using 16 steps with GtR, we achieve a 4.71× speedup (3.72s → 0.79s) compared to Harmon with 64 steps, while simultaneously improving the GenEval Overall score from 0.82 to 0.83. GtR maintains or improves performance across all sub-metrics, with particularly significant gains in Color Attribution (from 0.70 to 0.73). Under extreme settings (8 steps), GtR substantially outperforms the baseline (Overall score of 0.81 vs 0.74), further validating the effectiveness of GtR.
>
> **Rebuttal Table 2.1: GenEval performance comparison on Harmon at 512×512 resolution**
>
> | Method           | Latency(GPU/s)↓ | Speed↑ | Single Obj.↑ | Two Obj.↑ | Colors↑ | Counting↑ | Position↑ | Color Attri.↑ | Overall↑ |
> |------------------|----------------:|-------:|-------------:|----------:|--------:|----------:|----------:|--------------:|---------:|
> | steps=64         |            3.72 |   1.00 |         1.00 |      0.92 |    0.87 |      0.78 |      0.67 |          0.70 |     0.82 |
> | steps=32         |            2.17 |   1.71 |         1.00 |      0.93 |    0.88 |      0.76 |      0.64 |          0.70 |     0.82 |
> | **&nbsp;&nbsp;+GtR** |        **1.14** | **3.26** |     **0.99** |  **0.92** | **0.88** |  **0.80** |  **0.68** |      **0.73** | **0.83** |
> | steps=16         |            1.42 |   2.62 |         0.99 |      0.90 |    0.87 |      0.73 |      0.64 |          0.71 |     0.81 |
> | **&nbsp;&nbsp;+GtR** |        **0.79** | **4.71** |     **1.00** |  **0.92** | **0.88** |  **0.79** |  **0.67** |      **0.73** | **0.83** |
> | steps=8          |            1.06 |   3.51 |         0.99 |      0.85 |    0.84 |      0.60 |      0.60 |          0.57 |     0.74 |
> | **&nbsp;&nbsp;+GtR** |        **0.61** | **6.10** |     **0.99** |  **0.89** | **0.90** |  **0.74** |  **0.66** |      **0.69** | **0.81** |
>
> To further validate GtR's generalizability, we evaluated it on detail-dense prompts (images with rich textures but weak overall structure) and non-square layouts (512×384). As shown in Rebuttal Tables 2.2 and 2.3, GtR with 16 steps achieves 4.71× speedup while maintaining comparable generation quality, demonstrating strong generalizability across different content types and aspect ratios without model-specific tuning.
>
> **Rebuttal Table 2.2: Image Reward comparison on Harmon with detail-dense prompts and weak structure**
>
> | Method           | Latency(GPU/s)↓ | Speed↑ | Image Reward↑ |
> |------------------|----------------:|-------:|--------------:|
> | steps=64         |            3.72 |   1.00 |          0.90 |
> | steps=32         |            2.17 |   1.71 |          0.90 |
> | **&nbsp;&nbsp;+GtR** |        **1.14** | **3.26** |      **0.90** |
> | steps=16         |            1.42 |   2.62 |          0.86 |
> | **&nbsp;&nbsp;+GtR** |        **0.79** | **4.71** |      **0.90** |
> | steps=8          |            1.06 |   3.51 |          0.69 |
> | **&nbsp;&nbsp;+GtR** |        **0.61** | **6.10** |      **0.79** |
>
> **Rebuttal Table 2.3: Image Reward comparison on Harmon with non-square layouts (512×384)**
>
> | Method           | Latency(GPU/s)↓ | Speed↑ | Image Reward↑ |
> |------------------|----------------:|-------:|--------------:|
> | steps=64         |            3.72 |   1.00 |          0.82 |
> | steps=32         |            2.17 |   1.71 |          0.78 |
> | **&nbsp;&nbsp;+GtR** |        **1.14** | **3.26** |      **0.83** |
> | steps=16         |            1.42 |   2.62 |          0.76 |
> | **&nbsp;&nbsp;+GtR** |        **0.79** | **4.71** |      **0.82** |
> | steps=8          |            1.06 |   3.51 |          0.55 |
> | **&nbsp;&nbsp;+GtR** |        **0.61** | **6.10** |      **0.67** |

---

> ### Author Response · Authors · 2025-11-26
> **Response to reviewer  fAqu (2/N)**
>
> > **Q2.2.** The novelty of the work is limited. The core concept of establishing a global structure before refining details ("global-to-detail") has been explored in prior work like MUSE[1] and Hi-MAR[2] (missing reference and comparison). The paper's main contribution appears to be adapting this concept specifically to the sampling process of MAR.
>
> Thank you for this feedback. We agree that the global-to-detail paradigm has been explored in prior works such as Muse and Hi-MAR. However, GtR differs fundamentally in both mechanism and implementation.
>
> **Single-scale vs multi-scale token representation.** Hi-MAR and Muse adopt multi-scale image tokens, requiring low-resolution tokens as key-value caches to guide higher-resolution generation. This design significantly increases memory overhead and algorithmic complexity. In contrast, GtR simply reorders the sampling process of single-scale tokens without altering the token representation itself, ensuring minimal memory footprint and seamless compatibility with existing vision backbones and unified multimodal models.
>
> **Training-free with zero overhead.** Muse requires two VQGANs and two generative models, while Hi-MAR introduces two diffusion heads with scale-aware Transformer blocks and specialized multi-scale loss functions. Both methods demand substantial additional parameters and expensive, unstable training. GtR is entirely training-free and introduces no additional parameters or computational overhead, enabling immediate application to any pretrained MAR model.
>
> **Design principle.** While Hi-MAR and Muse achieve global-to-detail generation through architectural modifications for multi-resolution modeling, GtR stems from two observations about single-resolution generation (Figures 1 and 2): spatially adjacent tokens exhibit strong mutual influence during decoding, and uniformly distributed tokens across spatial locations determine image content while reconstruction of remaining tokens becomes substantially easier. These observations motivate our checkerboard sampling strategy, representing a conceptually distinct approach from multi-resolution methods.
>
> **Frequency-Weighted Token Selection (FTS).** GtR introduces FTS that exploits frequency domain information from diffusion features to identify tokens requiring enhanced diffusion steps. This differs fundamentally from Muse's confidence-based approach and is specifically tailored for continuous-valued generation in MAR models.
>
> **Extensive validation across scales.** We validated GtR on MAR, LightGen, and Harmon spanning 208M to 3.4B parameters across both class-conditional and text-to-image generation tasks, demonstrating strong scalability and generalizability without model-specific tuning.
>
> We will add Hi-MAR/MUSE to our comparisons and cite them appropriately in the revised paper.

---

> ### Author Response · Authors · 2025-11-26
> **Response to reviewer  fAqu (3/N)**
>
> > **Q2.3.** Insufficient Experiments: The paper only reports results for a very sparse set of sampling steps (e.g., steps 8 and 16 for C2I; steps 12 and 16 for T2I). It is unclear why step 8 was omitted for the T2I task, and the performance at other step counts remains unknown. The authors should provide the curves (e.g., FID vs. Sampling Steps) to show the full performance trend. Moreover, there are no trade-off curves to analyze the FID vs. speed when: (a) varying the step of the generation and reconstruction stages, (b) varying the autoregressive steps and the diffusion steps.
>
> Thank you for this suggestion. We have added the missing step 8 results on the T2I task and provide comprehensive trade-off curves to demonstrate the performance trends across different configurations.
>
> **(a) Step 8 on T2I task.** As shown in Rebuttal Table 2.4 below, compared to LightGen with 32 steps, GtR achieves a 4.47× speedup (from 1.03s to 0.23s) while maintaining nearly identical generation quality, with the GenEval Overall score decreasing by only 0.01 (from 0.55 to 0.54).
>
> **Rebuttal Table 2.4: GenEval Results on LightGen at 8 Sampling Steps**
>
> | Method           | Latency(GPU/s)↓ | Speed↑ | Single Obj.↑ | Two Obj.↑ | Colors↑ | Counting↑ | Position↑ | Color Attri.↑ | Overall↑ |
> |------------------|----------------:|-------:|-------------:|----------:|--------:|----------:|----------:|--------------:|---------:|
> | LightGen, 32     |            1.03 |   1.00 |         0.99 |      0.60 |    0.83 |      0.39 |      0.15 |          0.33 |     0.55 |
> | Step=8           |            0.57 |   1.81 |         0.98 |      0.55 |    0.87 |      0.39 |      0.12 |          0.21 |     0.52 |
> | **&nbsp;&nbsp;+GtR** |        **0.23** | **4.47** |     **1.00** |  **0.60** | **0.85** |  **0.38** |  **0.13** |      **0.30** | **0.54** |
>
> **(b) FID vs. Sampling Steps.** We provide the FID vs. Sampling Steps curves for both the original MAR and GtR in Figure 9 in our Appendix. The curves demonstrate that GtR consistently achieves superior generation quality compared to the original MAR across all sampling steps, with the advantage becoming more pronounced as the number of steps decreases.
>
> **(c) FID vs. speed when varying the steps of generation and reconstruction stages.** Figure 8(b) in our Appendix presents FID vs. Speed trade-off curves when varying the generation stage steps while fixing the reconstruction stage to either 1 or 2 steps. Regardless of whether the reconstruction stage uses 1 or 2 steps, GtR consistently outperforms the original MAR at equivalent speedup ratios. Notably, when the speedup ratio exceeds 3.2×, the original MAR exhibits severe quality degradation, whereas GtR maintains stable generation quality even at 8× speedup.
>
> **(d) FID vs. speed when varying the autoregressive steps and diffusion steps.** We evaluate the impact of initial diffusion steps $T_{\text{max}}$ on the quality-speed trade-off in Figure 8(a) in our Appendix. The curves show that GtR outperforms the original MAR for both $T_{\text{max}}=50$ and $T_{\text{max}}=100$ settings. While GtR with $T_{\text{max}}=100$ exhibits slightly lower performance compared to $T_{\text{max}}=50$, it still maintains substantial advantages over the original MAR baseline.
>
> These results demonstrate that GtR consistently outperforms the baseline across different sampling steps and hyperparameter settings.
>
> > **Q2.4.** Do not provide an introduction to the baseline methods (Halton, DiSA, LazyMAR).
>
> Thank you for this suggestion. We will provide descriptions of the baseline methods (Halton, DiSA, LazyMAR) in the revised paper.

---

### Official Review · Reviewer_tfUA · 2025-11-01

**Soundness:** 3
**Presentation:** 3
**Contribution:** 2
**Rating:** 6
**Confidence:** 4

**Summary:**

The authors identify some issues with MAR and related frameworks, particularly related to modeling potentially correlated tokens as independent when denoising them parallely. On this hypothesis, they suggest a two step generation methodology - first generate tokens in a checkerboard like pattern so as to avoid adjacent tokens being unmasked parallely, and once this provides sufficient context, reconstruct the rest of the tokens quickly in 1-2 steps, conditioned on the previously generated tokens. This leads to inference time accelerations as suggested by the empirical results, with minimal or no degradation.

**Strengths:**

- The authors present a comprehensive review of the literature in the context of visual generation, and identify shortcomings, proposing a sound method to address these limitations
- The two stage generation pipeline is a fairly novel contribution, utilizing ideas from previous works on token ordering
- The inference time speedups in terms of real time latency is promising
- The authors perform a detailed set of ablations, explaining their design choices

**Weaknesses:**

- The key hypothesis of this method is that the checkerboard-like pattern would lead to good global context and therefore more coherent generation. However, it is possible for images to have structures that extend to larger regions of the image, thus still potentially producing inconsistencies in generation. A deeper discussion on this point would be helpful
- Adding on to the previous point, since the checkerboard pattern followed by reconstruction is meant to produce more coherent generations, this approach should have led to generations with better fidelity, alongside latency speedups. However, the empirical results suggest neutral quality and only latency speedups when compared to corresponding baselines. It would be helpful to see the generations with highest FID to inspect and analyse the shortcomings of the proposed method

**Questions:**

Please see the weaknesses section above

---

> ### Author Response · Authors · 2025-11-26
> **Response to reviewer  tfUA (1/N)**
>
> > **Q1.1.** The key hypothesis of this method is that the checkerboard-like pattern would lead to good global context and therefore more coherent generation. However, it is possible for images to have structures that extend to larger regions of the image, thus still potentially producing inconsistencies in generation. A deeper discussion on this point would be helpful
>
> Thank you for raising this important point. We agree that structures extending to larger regions pose a challenging scenario. GtR offers advantages in handling such cases through its two-stage design. During the generation stage, each autoregressive step generates a small number of tokens, enabling bidirectional attention to effectively model inter-token dependencies. By the end of this stage, GtR ensures half the tokens are uniformly distributed across the structure, avoiding high entropy regions. In the reconstruction stage, generated tokens provide strong local conditioning for ungenerated tokens, enabling accurate reconstruction of extended structures.
>
> To validate this advantage, we evaluated GtR on Harmon using 100 prompts describing structures that extend to larger regions. These prompts feature large-scale compositional elements spanning substantial portions of the image, such as panoramic landscapes with rivers flowing from foreground to background, mountain ranges extending across the horizon, or architectural structures with bridges connecting distant regions.
>
> As shown in Rebuttal Table 1.1 below, GtR with 32 steps achieves a 3.26× speedup compared to Harmon with 64 steps while maintaining comparable generation quality (Image Reward: 1.18 vs 1.19). Under extreme settings (8 steps), GtR improves Image Reward by 0.14 (0.88 → 1.02) compared to original Harmon. These results demonstrate that GtR effectively handles prompts with structures extending to larger regions while providing substantial acceleration. We appreciate your suggestion and will include this analysis in the revised paper.
>
> **Rebuttal Table 1.1: Image Reward comparison on prompts with structures extending to larger regions**
>
> | Method           | Latency(GPU/s)↓ | Speed↑ | Image Reward↑ |
> |------------------|----------------:|-------:|--------------:|
> | steps=64         |            3.72 |   1.00 |          1.19 |
> | steps=32         |            2.17 |   1.71 |          1.15 |
> | &nbsp;&nbsp;+GtR |            1.14 |   3.26 |          1.18 |
> | steps=16         |            1.42 |   2.62 |          1.10 |
> | &nbsp;&nbsp;+GtR |            0.79 |   4.71 |          1.15 |
> | steps=8          |            1.06 |   3.51 |          0.88 |
> | &nbsp;&nbsp;+GtR |            0.61 |   6.10 |          1.02 |

---

> ### Author Response · Authors · 2025-11-26
> **Response to reviewer  tfUA (2/N)**
>
> > **Q1.2.** Adding on to the previous point, since the checkerboard pattern followed by reconstruction is meant to produce more coherent generations, this approach should have led to generations with better fidelity, alongside latency speedups. However, the empirical results suggest neutral quality and only latency speedups when compared to corresponding baselines. It would be helpful to see the generations with highest FID to inspect and analyse the shortcomings of the proposed method
>
> Thank you for this feedback. We would like to clarify that GtR achieves substantial speedups while simultaneously improving generation quality across multiple settings.
>
> As shown in Rebuttal Table 1.2, GtR with 40 steps achieves a 3.24× speedup compared to original MAR with 64 steps while improving both FID (1.59 → 1.58) and IS (299.1 → 304.28). To further demonstrate this phenomenon, we conducted additional experiments on Harmon, a more advanced MAR-based text-to-image model. As shown in Rebuttal Table 1.3, GtR with 16 steps achieves a 4.71× speedup compared to Harmon with 64 steps while improving the Overall GenEval score from 0.82 to 0.83, with particularly significant gains in Color Attribution from 0.70 to 0.73. These results demonstrate that the checkerboard generation followed by reconstruction not only accelerates sampling but also improves fidelity in practice.
>
> Regarding generations with highest FID, through qualitative inspection, we observe that cases with highest FID under GtR correspond to the same prompts where the baseline model without acceleration also produces poor generation quality. This suggests that these challenging cases represent limitations of the underlying generative model capacity. We will continue investigating these challenging cases to identify potential directions for further improving GtR's robustness.
>
> **Rebuttal Table 1.2: Performance comparison on MAR at different sampling steps**
>
> | Method        | Latency(GPU/s)↓ | Speed↑ | FID↓ |   IS↑ |
> |---------------|----------------:|-------:|-----:|------:|
> | MAR, steps=64 |            0.81 |   1.00 | 1.59 | 299.1 |
> | **GtR, steps=40** |            **0.25** |   **3.24** | **1.58** | **304.28** |
>
> **Rebuttal Table 1.3: GenEval performance comparison on Harmon at 512×512 resolution**
>
> | Method           | Latency(GPU/s)↓ | Speed↑ | Single Obj.↑ | Two Obj.↑ | Colors↑ | Counting↑ | Position↑ | Color Attri.↑ | Overall↑ |
> |------------------|----------------:|-------:|-------------:|----------:|--------:|----------:|----------:|--------------:|---------:|
> | steps=64         |            3.72 |   1.00 |         1.00 |      0.92 |    0.87 |      0.78 |      0.67 |          0.70 |     0.82 |
> | steps=32         |            2.17 |   1.71 |         1.00 |      0.93 |    0.88 |      0.76 |      0.64 |          0.70 |     0.82 |
> | **&nbsp;&nbsp;+GtR** |        **1.14** | **3.26** |     **0.99** |  **0.92** | **0.88** |  **0.80** |  **0.68** |      **0.73** | **0.83** |
> | steps=16         |            1.42 |   2.62 |         0.99 |      0.90 |    0.87 |      0.73 |      0.64 |          0.71 |     0.81 |
> | **&nbsp;&nbsp;+GtR** |        **0.79** | **4.71** |     **1.00** |  **0.92** | **0.88** |  **0.79** |  **0.67** |      **0.73** | **0.83** |
> | steps=8          |            1.06 |   3.51 |         0.99 |      0.85 |    0.84 |      0.60 |      0.60 |          0.57 |     0.74 |
> | **&nbsp;&nbsp;+GtR** |        **0.61** | **6.10** |     **0.99** |  **0.89** | **0.90** |  **0.74** |  **0.66** |      **0.69** | **0.81** |

---

### Author Response · Authors · 2025-12-03
**Summary of Rebuttal**

Thank you again to the reviewers for their thoughtful and constructive feedback, and to the (new) AC for taking on this additional responsibility in light of the recent OpenReview incident.

We are grateful for the positive feedback, reflected in scores **(8, 6, 6, 4)** and recognition across three key dimensions:

-   **Method:** "Conceptually novel... training-free hierarchical sampling strategy" (`Reviewer ZCm1`); "Sound method to address these limitations" (`Reviewer tfUA`); "Simple, practical idea with strong engineering value" (`Reviewer FhEK`); "Effective and intuitively easy to understand" (`Reviewer fAqu`).

-   **Evaluation:** "Detailed set of ablations, explaining their design choices" (`Reviewer tfUA`); "Extensive experiments... demonstrate that GtR achieves significant speedups" (`Reviewer ZCm1`); "The reported ≈3.72× acceleration while maintaining generation quality is compelling evidence of practical value" (`Reviewer FhEK`); "The performance in ImageNet is impressive" (`Reviewer fAqu`).

-   **Significance:** "The inference time speedups in terms of real time latency is promising" (`Reviewer tfUA`); "Better balance between generation speed and generation quality" (`Reviewer ZCm1`); "Immediately usable in deployed MAR systems... requires no model retraining" (`Reviewer FhEK`).

**Resolution of Key Concerns:**

We have carefully studied all comments and have provided detailed point-by-point responses under each Reviewer's section. To address the reviewers' concerns, we conducted extensive additional experiments, presenting **14 new tables** and **3 new figures** to demonstrate GtR's generalizability and robustness across diverse scenarios:

-   **Generalizability** (`Reviewers tfUA, fAqu, FhEK`): To validate GtR's generalizability, we conducted additional experiments on Harmon, a more advanced MAR-based text-to-image model (**Rebuttal Tables 1.3**). We evaluated GtR across multiple challenging scenarios including structures extending to larger regions (**Rebuttal Table 1.1**), detail-dense images with weak global structure (**Rebuttal Table 2.2**), and non-square layouts at 512×384 resolution (**Rebuttal Table 2.3**). Across these diverse scenarios, GtR consistently achieves **3.26×–6.10×** speedup while maintaining or improving generation quality. For example, on detail-dense prompts, GtR at 16 steps achieves 4.71× speedup compared to Harmon with 64 steps while maintaining comparable generation quality (Image Reward: 0.90 vs 0.90); on structures extending to larger regions, under extreme settings (8 steps), GtR improves Image Reward by 0.14 (0.88 → 1.02) compared to original Harmon. These results demonstrate strong generalizability without task-specific tuning.

-   **Theoretical Justification** (`Reviewer FhEK`): We provide comprehensive theoretical analysis grounded in information theory in **Appendix A.4**, proving that the checkerboard partition is near-optimal by formulating it as a Maximum Independent Set problem on grid graphs and exploiting local Markov properties.

-   **Robustness of Hyperparameters and Additional Experiments** (`Reviewers fAqu, FhEK, ZCm1`): We conducted systematic analyses demonstrating GtR's robustness. We added the step 8 results on T2I model LightGen as requested in Q2.3 (**Rebuttal Table 2.4**), FID vs. Sampling Steps curves (**Figure 9** in the revised paper), and FID vs. Speed trade-off curves under various settings (**Figure 8** in the revised paper) showing GtR consistently outperforms baselines across diverse configurations. Detailed latency breakdowns on multiple hardware platforms (RTX 4090, A100) validate hardware-agnostic efficiency gains (**Rebuttal Tables 3.3, 3.4**). Sensitivity analyses for key hyperparameters demonstrate stable performance across reasonable ranges: number of stages $K$ (**Rebuttal Tables 3.5-3.6**), FTS percentile $\beta$ (**Rebuttal Table 4.5**, **Figure 7** in the revised paper), and detailed diffusion steps $T_{\text{detail}}$ (**Rebuttal Table 3.7**). Comprehensive ablation studies on checkerboard pattern selection (**Rebuttal Table 4.4**) validate the robustness of GtR's strategy regardless of pattern choice. Our default settings achieve near-optimal results across multiple models (Harmon, MAR, LightGen) and diverse scenarios without extensive tuning.

-   **Distinctive Contributions** (`Reviewer fAqu`): We clarified GtR's distinctions from prior work (Hi-MAR, Muse) in **Q2.2**, emphasizing our single-scale token representation, training-free design with zero overhead, and Frequency-Weighted Token Selection (FTS) specifically tailored for continuous-valued MAR models.

In conclusion, the reviewers show a broadly positive view of the paper, recognizing it as a "conceptually novel" and "practical" solution with "strong engineering value." We believe we have fully addressed the remaining questions, and we hope this work offers valuable insights to the visual generation community

Best regards,

**Authors of Submission 12483**

---

### Meta-Review · Area_Chair_or31 · 2025-12-29

**Summary:**

All reviewers acknowledged GtR’s novel, training-free two-stage sampling strategy for MAR models, noting its 3.72× speedup while maintaining generation quality and strong engineering value. Key concerns included limited generalizability across tasks and layouts, unclear theoretical support for checkerboard sampling, insufficient hyperparameter robustness checks, and vague step allocation details. The authors fully resolved these issues with extensive supplementary experiments, theoretical analysis, and clarity improvements, thus I would support the paper’s acceptance.

**Reviewer Concerns:**

All key issues were resolved. The authors validated GtR’s generalizability on detail-dense images, non-square layouts, and advanced T2I models; provided information-theoretic proof for checkerboard sampling optimality; conducted hardware-agnostic latency breakdowns; and performed rigorous hyperparameter sensitivity tests showing stable performance. They also clarified step allocation, baseline descriptions, and method distinctions from prior work like MUSE/Hi-MAR.

**Reviewer Scores:**

Currently most of the reviewers have positive ratings. The negative reviewer also mentioned the willingness to raise the score if the concerns are well addressed. Overall, I think the overall rating will be positive of all reviewers had been able to fully participate in the discussion.

---

### Decision · Program_Chairs · 2026-01-26

Accept (Poster)